# Quantitative regulation of the dynamic steady state of actin networks

**Angelika Manhart[1,2†]\*, Téa Aleksandra Icheva[3†], Christophe Guerin[3], Tobbias Klar[3], Rajaa Boujemaa-Paterski[3§], Manuel Thery[3,4], Laurent Blanchoin[3,4‡]\*, Alex Mogilner[1,2‡]\***

[1]Courant Institute of Mathematical Sciences, New York University, New York, United States; [2]Department of Biology, New York University, New York, United States; [3]CytomorphoLab, Biosciences & Biotechnology Institute of Grenoble, Laboratoire de Physiologie Cellulaire & Végétale, Université Grenoble-Alpes/CEA/CNRS/INRA, Grenoble, France; [4]CytomorphoLab, Hôpital Saint Louis, Institut Universitaire d'Hematologie, UMRS1160, INSERM/AP-HP/Université Paris Diderot, Paris, France

**Abstract** Principles of regulation of actin network dimensions are fundamentally important for cell functions, yet remain unclear. Using both in vitro and in silico approaches, we studied the effect of key parameters, such as actin density, ADF/Cofilin concentration and network width on the network length. In the presence of ADF/Cofilin, networks reached equilibrium and became treadmilling. At the trailing edge, the network disintegrated into large fragments. A mathematical model predicts the network length as a function of width, actin and ADF/Cofilin concentrations. Local depletion of ADF/Cofilin by binding to actin is significant, leading to wider networks growing longer. A single rate of breaking network nodes, proportional to ADF/Cofilin density and inversely proportional to the square of the actin density, can account for the disassembly dynamics. Selective disassembly of heterogeneous networks by ADF/Cofilin controls steering during motility. Our results establish general principles on how the dynamic steady state of actin network emerges from biochemical and structural feedbacks.
DOI: https://doi.org/10.7554/eLife.42413.001

**\*For correspondence:**
a.manhart@imperial.ac.uk (AM);
laurent.blanchoin@cea.fr (LB);
mogilner@cims.nyu.edu (AM)

[†]These authors contributed equally to this work
[‡]These authors also contributed equally to this work

**Present address:** [§]Department of Biochemistry, University of Zurich, Zurich, Switzerland

**Competing interests:** The authors declare that no competing interests exist.

## Introduction

Dynamic actin networks play important roles in cell migration (*Rottner and Stradal, 2011*), morphogenesis (*Hopmann and Miller, 2003*), immune response (*Vargas et al., 2016*) and intracellular pathogen motility (*Reed et al., 2014*). The architecture and geometry of the actin networks are tightly controlled in these essential cellular processes, and defects in this control cause pathologies, such as ageing disorders (*Amberg et al., 2011*). Here, we focus on the steady state dynamics of branched filament arrays that are initiated by the Arp2/3 complex (*Rotty et al., 2013*) and activated by WASP family proteins (*Krause and Gautreau, 2014*), which are instrumental in lamellipodial extension (*Krause and Gautreau, 2014*), pathogen propulsion (*Reed et al., 2014*), endo- and exocytosis (*Li et al., 2018*).

In many cellular processes, the branched actin networks are polarized and appear in a state of dynamic equilibrium: at their leading edge, barbed filament ends are oriented forward and polymerize, elongating the network, while throughout the network a net disassembly takes place, gradually thinning the network out and limiting the network to a finite equilibrium length. As a result, the network exists in a 'global treadmilling state' (*Borisy and Svitkina, 2000*; *Pollard and Borisy, 2003*; *Carlier and Shekhar, 2017*; *Koestler et al., 2013*) – as opposed to the well-characterized treadmilling of individual filaments. Important examples of such networks are flat lamellipodia at the leading

edge of cells migrating on flat surfaces (*Rottner and Stradal, 2011*; *Barnhart et al., 2011*; *Ofer et al., 2011*; *Raz-Ben Aroush et al., 2017*; *Rottner and Schaks, 2019*) and in 3D extracellular matrix (*Fritz-Laylin et al., 2017*) and cylindrical actin tails propelling intracellular pathogens (*Theriot et al., 1992*; *Rosenblatt et al., 1997*; *Lacayo et al., 2012*; *Reed et al., 2014*), endosomes and lysosomes (*Taunton et al., 2000*).

In what follows, we call the distance from the leading to trailing edge as network length, and the characteristic dimension of the leading edge – network width. Both the length and width of the dynamic network are important physiological parameters (*Carlier and Shekhar, 2017*) that have to be regulated. For example, the width of the actin tails is usually approximately equal to the size of the pathogen or organelle, which is being propelled by the tail, and the length, presumably, has to be sufficient for the tail to be enmeshed with the host cell cytoskeletal scaffold. The width and length of lamellipodia probably have to be sufficient to fit into the geometry of the extracellular matrix and to accommodate other cytoskeletal elements, such as stress fibers.

Assembly and disassembly play central roles in determining actin network length (*Theriot et al., 1992*; *Ofer et al., 2011*). Yet, while assembly is relatively well studied (*Rottner and Schaks, 2019*), systems-level understanding of disassembly is lacking. In keratocytes' cytoplasmic fragments, the lamellipodial length, $L$, is simply determined by the time necessary for the disassembly, characterized by rate $1/\tau$ to largely degrade the lamellipodial network assembled at the leading edge. So, if the actin network growth rate is $V$, then $L \sim V\tau$ (*Ofer et al., 2011*). Similarly, in *Listeria*'s actin tail, the network density decreases exponentially, with a constant rate, and the tail's length is proportional to the pathogen's speed (*Theriot et al., 1992*).

As demonstrated both in vivo and in vitro, proteins of the ADF/Cofilin family play a key role in the actin disassembly (*Bamburg, 1999*), debranching the network, severing the filaments (*Blanchoin et al., 2014*) and accelerating depolymerization at filaments' ends (reviewed in *Carlier and Shekhar (2017)*). Microscopic details of the ADF/Cofilin-mediated kinetics of actin filaments at the molecular level are being clarified (*Wioland et al., 2017*), but so far there is little understanding about how the net rate of the network disassembly, rather than that of individual filaments, is determined by the geometry and architecture of the network and by the actin and ADF/Cofilin concentrations. Furthermore, spatio-temporal dynamics of ADF/Cofilin and its relation to the network disassembly remains obscure. Lastly, actin-network steering, linked to the regulation of network growth at the leading edge (*Boujemaa-Paterski et al., 2017*), is essential to understanding directional cell motility. However, how organization and dynamics of the bulk of the actin network affects the steering is unclear (*Krause and Gautreau, 2014*). In this study, we investigated how the geometry, architecture and density of a branched actin network, as well as the ADF/Cofilin concentration, affect the actin network dynamics, and found key parameters controlling the network length and steering.

In order to do that, we combined in vitro and in silico, approaches. In vitro reconstitution of bacteria and plastic beads propulsion (*Frischknecht et al., 1999*; *Loisel et al., 1999*; *Bernheim-Groswasser et al., 2002*; *Akin and Mullins, 2008*; *Dayel et al., 2009*; *Achard et al., 2010*; *Kawska et al., 2012*), and of lamellipodial network growth (*Bieling et al., 2016*; *Boujemaa-Paterski et al., 2017*) brought insights on how a minimal set of just two molecular actions – Arp2/3 complex-driven nucleation and barbed-end capping – can result in the actin leading edge organization and growth. In this study, we added ADF/Cofilin to the mixture of actin, Arp2/3 complex and capping protein in an experimental chamber with the nucleation promoting factor (NPF) Human WASp-pVCA, localized to micro-printed patterns on the surface. We generated a diversity of patterns and studied the impact of the geometry and actin density on the length of dynamic actin networks. We also used quantitative fluorescence imaging to measure the spatial and temporal behavior of the actin and ADF/Cofilin densities and their relations with the network length. Crucially, we varied independently three parameters – actin network density, ADF/Cofilin concentration and network width – and measured their effect on the network length.

Mathematical modeling was very successful in deciphering the data from in vitro experiments on the actin disassembly (*Roland et al., 2008*; *Berro et al., 2010*; *Michalski and Carlsson, 2010*; *Michalski and Carlsson, 2011*; *Reymann et al., 2011*; *Stuhrmann et al., 2011*; *Tania et al., 2013*). Most theoretical studies either considered the disassembly of individual filaments (*Roland et al., 2008*), or a first-order reaction of a continuous network density decrease (*Ofer et al., 2011*; *Reymann et al., 2011*; *Stuhrmann et al., 2011*), or treated the disassembly as a boundary condition

(*Raz-Ben Aroush et al., 2017*). Pioneering theory of *Michalski and Carlsson (2010)*, *Michalski and Carlsson (2011)* demonstrated how fragmentation of the network at the trailing edge resulted from stochastic accumulation of discrete disassembly events in the network. No studies so far quantitatively connected the dynamics of ADF/Cofilin accumulation in the actin mesh with the effective disassembly rate and the network length.

An intuitive and expected qualitative finding of our study is that equilibrium network length increases with actin density, and decreases with ADF/Cofilin concentration. The main insight of the study is quantitative: we found a novel, simple mathematical relation allowing the prediction of the actin network length from three parameters – actin network density, ADF/Cofilin concentration and network width – and measured their effect on the network length. Other novel findings are: 1) ADF/Cofilin is locally depleted from solution by binding to actin, which has profound effects on actin disassembly; 2) Network length depends on the network width; 3) ADF/Cofilin concentration can regulate the steering of heterogeneous actin networks.

## Results

### ADF/Cofilin action establishes equilibrium length of dynamic actin networks

We reconstituted branched actin networks (called LMs in the following) that resemble lamellipodia of motile cells by micro-printing rectangular patterns coated with nucleation-promoting factors (NPFs) on the 'bottom' of the experimental chamber. NPFs activated the Arp2/3 complex, which in turn generated filament branching, leading to the assembly and growth of the branched actin network at the rectangular network leading edge pattern (*Figure 1A*, *Figure 1—figure supplement 1A*). The thickness of the experimental chamber ('bottom-to-top distance') is only a few microns, so the actin networks lift off the NPF pattern, bump into the 'top', bend and then grow parallel to the bottom and top (*Figure 1—figure supplement 1B*). The networks were flat, similar to the lamellipodial networks: their thickness was but a few microns, an order of magnitude less than the width and length, on the order of tens of microns. Importantly, there is capping protein in the reaction mixture, limiting growth of individual actin filaments and keeping the actin networks compact, not extending laterally from the NPF pattern. With only actin, Arp2/3 complex and capping protein in the reaction mixture, the networks elongated steadily (*Figure 1A*). The networks' elongation speed *V* was an increasing function of actin density (*Figure 1B–C*), in agreement with our previous study *Boujemaa-Paterski et al. (2017)*. In *Figure 1—figure supplement 1D* and *Figure 1—video 1*, we report data suggesting that the higher NPF density both increases the actin density, and translates polymerization into the network elongation more effectively, without changing the rate of filament growth.

Without ADF/Cofilin, the networks elongated steadily and did not disassemble – actin density along the networks changed only slightly (*Figure 1A*). Addition of ADF/Cofilin changed the networks' dynamics: rather than growing steadily, the networks, after reaching a certain length, started to disassemble at the trailing edge, so that a dynamic steady state was reached in which the network length stayed roughly constant (*Figure 2A*, *Figure 2—figure supplement 1*). The equilibrium length depended on both actin density and ADF/Cofilin concentration. The addition of ADF/Cofilin did not have a significant effect on the growth rate of the networks, in contrast to the in vivo cases. The reason is that the total amount of actin in the in vitro chamber is vastly greater than the total network actin, and so the polymerizable actin monomer concentration is unaffected by the actin turnover related to the networks' dynamics; in other words, actin does not have to be recycled. This has an important consequence for the in vitro global treadmill: the rate of the network growth depends on the conditions at the leading edge (actin density and architecture) but is unaffected by the network length. Thus, the equilibrium length of the treadmilling network is determined by the length-dependent disassembly only: the longer the network is, the faster is the disassembly at the trailing edge, and so the treadmilling length is determined by the dynamic stable equilibrium, in which the trailing edge disassembly rate is equal to the leading-edge growth rate. As the leading-edge growth rate is unaffected by ADF/Cofilin, our in vitro assay allows investigation of the effect of the ADF/Cofilin-mediated disassembly on the network length, without complications of feedbacks between disassembly and assembly.

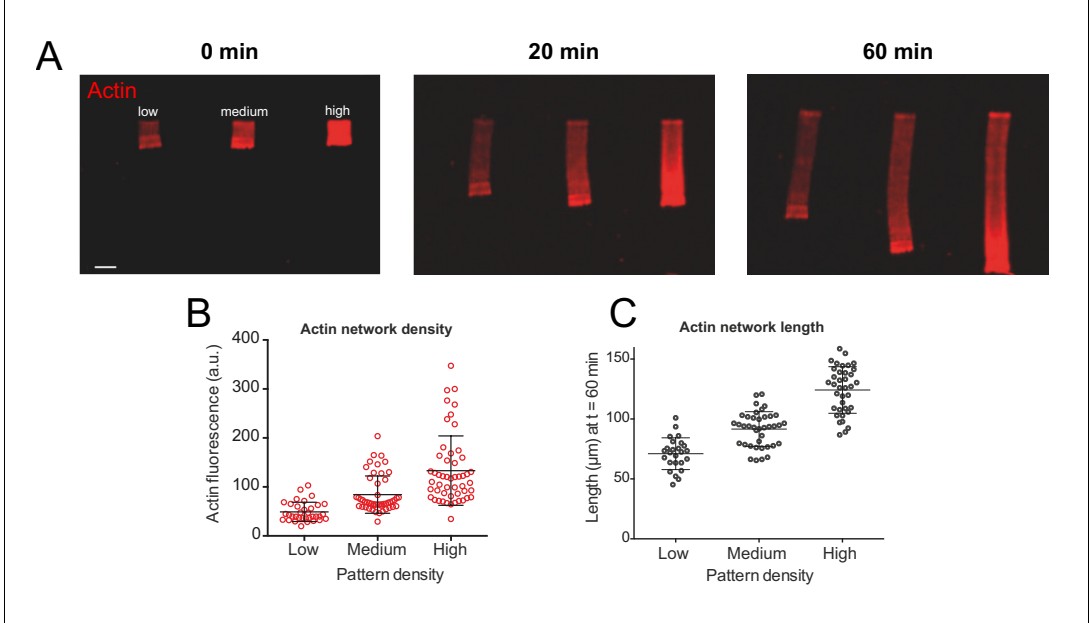

**Figure 1.** Actin density determines network growth speed. (**A**) The growth kinetics of reconstituted lamellipodia (LMs) depend on the density of the grafted NPFs (Human WASp-pVCA). Conditions: 6 μM actin monomers Alexa-568 labeled, 18 μM profilin, 120 nM Arp2/3 complex, 20 nM capping proteins. LMs of variable actin filament density (low, medium, high) were initiated by bar-shaped patterns of different NPF-spot densities (see *Figure 1—figure supplement 1*) and their growth was followed using the fluorescence of the actin networks. Snapshots of the growing lamellipodium were taken 0, 20 and 60 min after addition of Alexa-568-labeled actin monomers. (**B**) Denser patterns generate denser actin networks. The network density was measured across the LMs (for low density networks, n = 25 from three experiments, for medium-density networks, n = 41 from three experiments and for high-density networks, n = 38 from three experiments). (**C**) Denser patterns generate longer actin networks. The lengths of the LMs were measured after 60 min and plotted according to the pattern density.

DOI: https://doi.org/10.7554/eLife.42413.002

The following video and figure supplement are available for figure 1:

**Figure supplement 1.** Laser micropatterning method and growth rates.

DOI: https://doi.org/10.7554/eLife.42413.003

**Figure 1—video 1.** Branched actin filaments growing on a laser-patterned surface.

DOI: https://doi.org/10.7554/eLife.42413.004

Our data revealed that the equilibrium network length decreases with the ADF/Cofilin concentration and increases with the actin density (*Figure 2A–B*). Qualitatively, these results are very intuitive: higher ADF/Cofilin concentration increases the disassembly rate, hence the equilibrium between the leading edge growth and trailing edge disassembly is reached at shorter lengths. If the actin network is denser at the leading edge, it takes a longer time to break such network down; during this time, the steadily elongating network grows longer until the disassembly rate at the trailing edge balances the leading edge growth.

## Spatio-temporal ADF/Cofilin dynamics and its local depletion
### Initial simple model of the ADF/Cofilin dynamics

We wondered if these observations could be explained by a simple model: ADF/Cofilin binds to every spot of the growing actin network with a constant rate and does not have an effect on the network until a critical density of the bound ADF/Cofilin is reached, upon which the network disassembles instantly. It is reasonable to assume that ADF/Cofilin binding is a diffusion-limited reaction, and so its rate is proportional to the product of the ADF/Cofilin concentration in the solution, $C_0$, and of the constant actin filament density, $A$. Indeed, when we analyzed the initial rate of binding of ADF/Cofilin near the leading edge for networks that had just started to grow (using various actin densities and at various ADF/Cofilin concentrations), we found that this rate is proportional to $C_0$. We also found a strong correlation between the initial increase in bound ADF/Cofilin and the product $C_0 \times A$

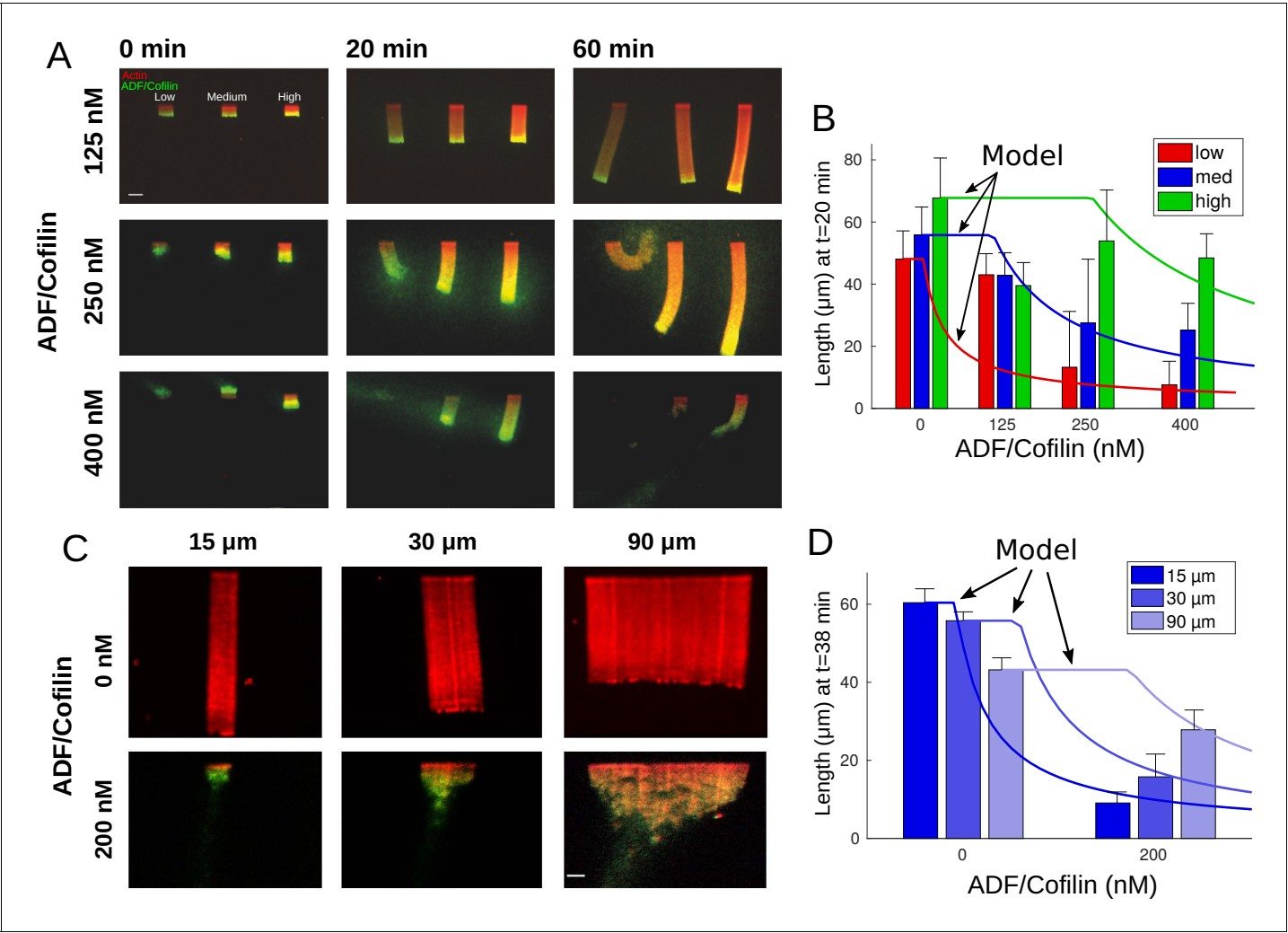

**Figure 2.** Actin network density and width set the equilibrium state of LMs. (**A**) The growth kinetics of LMs in the presence of ADF/Cofilin. The experiment conditions are similar to *Figure 1A* but with addition of variable concentrations of ADF/Cofilin as indicated. Snapshots of the growing lamellipodium were taken 0, 20 and 60 min after addition of actin monomers. Actin monomers are Alexa-568 labeled (red), ADF/Cofilin is labeled with Alexa-488 (green). Scale bar is 15 μm. (**B**) The length of the actin networks as a function of the ADF/Cofilin concentrations were measured after 20 min. Colored bars (red for low, blue for medium and green for high actin density) at 20 min are averages (± standard deviations). Solid lines with corresponding colors are the model prediction according to Sec. Equilibrium length of actin network as a function of biochemical and geometric parameters. The horizontal lines for the model predictions indicate that the networks have not yet reached equilibrium (0 nM ADF/Cofilin, n = 33 for low, n = 41 for medium, n = 38 for high from three experiments; 125 nM ADF/Cofilin, n = 15 for low, medium and high from three experiments; 250 nM ADF/Cofilin, n = 26 for low, n = 27 for medium and n = 27 for high from three experiments; 400 nM ADF/Cofilin, n = 19 for low, medium and high from three experiments). (**C**) Growth of LMs from patterns of different sizes. Biochemical conditions are identical to *Figure 2A* and *Figure 1A*. Top panel, LMs in the absence of ADF/Cofilin were initiated from pattern of three different sizes (15 × 3, 30 × 3 and 90 × 3 μm²). Snapshots were taken at 36 min after addition of actin monomers (15 μm n = 50 from three experiments, 30 μm n = 59 from 11 experiments and 90 μm, n = 43 from 10 experiments). See *Figure 2—video 1* for full dynamics. Bottom panel, LMs in presence of 200 nM ADF/Cofilin were initiated from pattern of three different sizes (15 × 3, 30 × 3 and 90 × 3 μm²). Snapshots were taken at 38 min after addition of actin monomers. Scale bar is 15 μm. See *Figure 2—video 2* for the full dynamics. (**D**) Measured actin network lengths as a function of ADF/Cofilin concentration. Colored bars are the average length (±38 standard deviation) min after the addition of actin monomers. Lines show the model prediction of Sec. Equilibrium length of actin network as a function of biochemical and geometric parameters. Horizontal lines for the model predictions indicate that the networks have not yet reached equilibrium.

DOI: https://doi.org/10.7554/eLife.42413.005

The following video and figure supplement are available for figure 2:

**Figure supplement 1.** Growth of networks of varying widths.

DOI: https://doi.org/10.7554/eLife.42413.006

**Figure 2—video 1.** Growth of LMs from pattern of different size in absence of ADF/Cofilin.

DOI: https://doi.org/10.7554/eLife.42413.007

*Figure 2 continued on next page*

*Figure 2 continued*
**Figure 2—video 2.** Growth of LMs from pattern of different size in presence of ADF/Cofilin.
DOI: https://doi.org/10.7554/eLife.42413.008

($R = 0.51, P<0.001$, see Appendix 1 for details). This confirms that at least at the beginning of network growth, the ADF/Cofilin binding rate is indeed $k_B C_0 A$, where $k_B$ is the binding constant.

If this rate stays constant, then the bound ADF/Cofilin density as a function of time and of distance $y$ from the network leading edge is the solution of the equation $\partial_t C_B + V \partial_y C_B = k_B C_0 A$, where $V$ is the rate of actin network growth at the leading edge. Since newly polymerized actin is free of ADF/Cofilin, we can assume $C_B(y = 0) = 0$. In dynamic equilibrium, this equation yields the solution $C_B(y,t) = \frac{k_B C_0 A y}{V}$, which can be easily understood: an actin spot takes time $y/V$ to drift a distance $y$ from the leading edge. As ADF/Cofilin binds with rate $k_B C_0 A$, by that time the bound ADF/Cofilin density reaches the value of $\frac{k_B C_0 A y}{V}$. Assuming that the network falls apart when a critical amount of ADF/Cofilin per actin filament, $C_B/A = \gamma$, is reached, this yields an equilibrium network length of

$$L_* = \frac{\gamma V}{k_B C_0}.$$

This simple model predicts that the equilibrium network length is proportional to the ADF/Cofilin concentration in the solution, $C_0$, in qualitative agreement with the data (compare *Figure 2A–B*). In *Figure 2A–B*, we also observe a clear correlation between the actin density and the network length. Since denser networks also grow faster (*Figure 1C*), our estimate is again in qualitative agreement with *Figure 2A–B*, however, it appears that the network growth rate increases only weakly with the actin density, while the equilibrium network length increases dramatically, when the actin filament density increases. Lastly, the simple model indicates that the equilibrium network length is independent of the network width.

## Equilibrium network length increases with the network width

We tested this last prediction experimentally for networks of widths 15, 30 and 90 µm, and the result shows that this is not the case (*Figure 2C–D*, *Figure 2—video 1*, *Figure 2—video 2*). In fact, we observed that, while for all network widths their lengths decrease if ADF/Cofilin is added, wider networks are affected less. This suggests three potential factors that the simple initial model did not take into account: (1) ADF/Cofilin is unable to diffuse from the solution to the inner parts of the wider dense actin network. (2) There is a non-local mechanical effect that leads to an effective protection of wider networks against degradation. (3) Local depletion of ADF/Cofilin. As previously reported (*Boujemaa-Paterski et al., 2017*), actin monomers are locally depleted due to a sink of its concentration in the vicinity of the growing barbed ends; a similar effect could emerge for ADF/Cofilin.

To estimate the potential effect of the actin network on the ADF/Cofilin diffusion constant, we used the theory developed in *Novak et al. (2009)* and described in Appendix 1 to determine the effective diffusion constant of ADF/Cofilin inside the actin network. This calculation shows that the effect of even a dense actin network on the ADF/Cofilin diffusion coefficient is a reduction by a few percent only, that is the diffusion constant will be virtually unaffected by the actin network, ruling out the first factor. The second factor, a global mechanical structure of the network, is unlikely, since the average actin filaments are of sub-micron size, two orders of magnitude shorter than the network width, and long actin bundles are absent. Thus, we decided to investigate the third factor, local depletion of ADF/Cofilin.

## Rate of ADF/Cofilin binding decreases with time

According to the simple initial model, the rate of ADF/Cofilin binding to an actin spot, $k_B C_0 A$, should be constant, not changing with time. If we focus on such a spot drifting from the leading edge, we should measure a linear increase of the ADF/Cofilin density with the slope that does not depend on the time when the spot originates. We examined such an increase of the ADF/Cofilin density near the leading edge by making measurements at different starting times. We indeed found that the

increase of the ADF/Cofilin density is linear with time; however, the rate of the increase decreased with starting time (*Figure 3A–B*), rather than remaining constant.

## ADF/Cofilin is locally depleted by binding to the growing actin network

To confirm that the observed decrease of the ADF/Cofilin binding rate with time is due to the local ADF/Cofilin depletion, we analyzed the simplest model of the spatial-temporal ADF/Cofilin dynamics compatible with our observation. In the model, the densities of free ADF/Cofilin molecules diffusing in the solute and of ADF/Cofilin molecules bound to the network are $C_F(x, y, t)$ and $C_B(x, y, t)$,

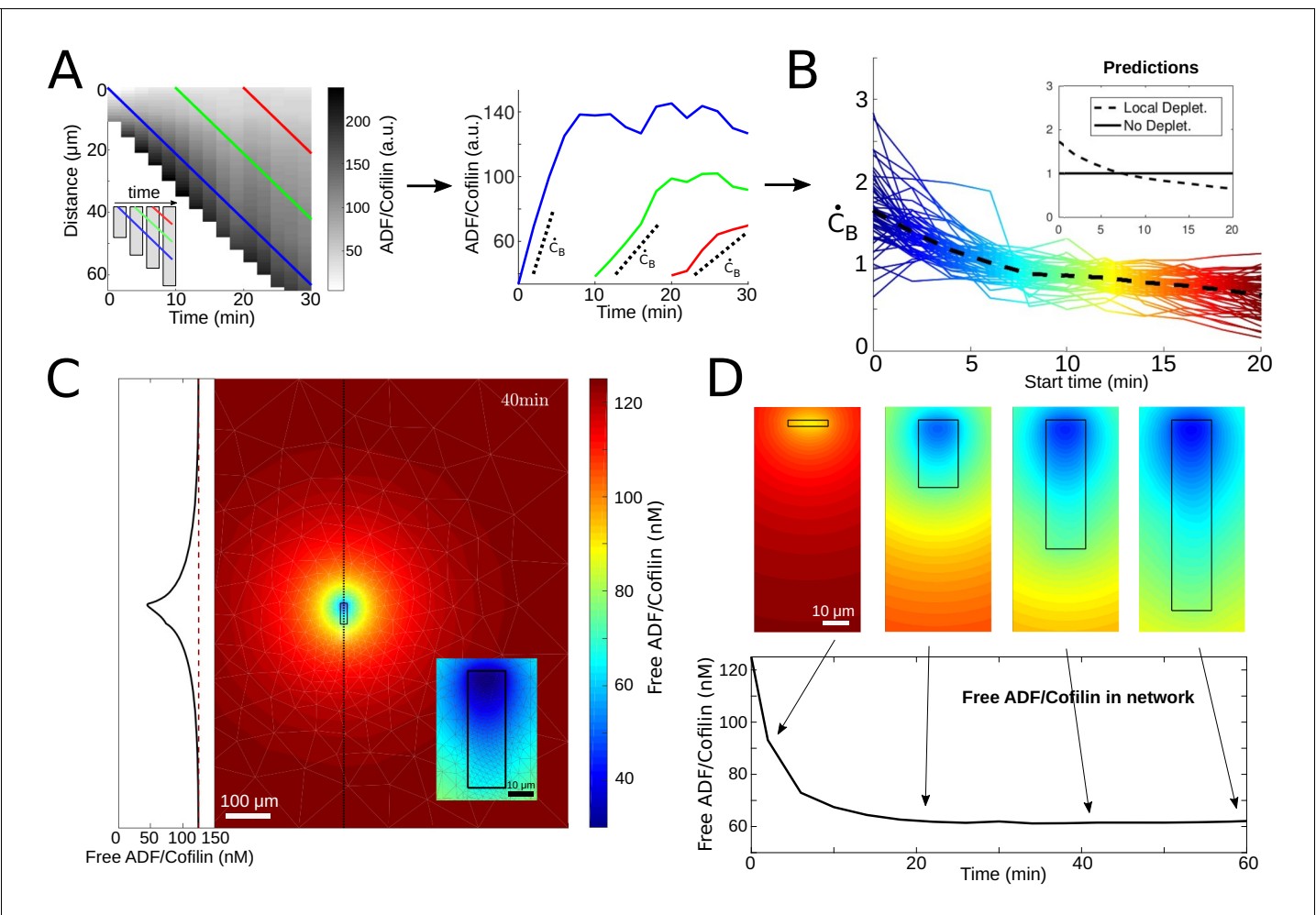

**Figure 3.** ADF/Cofilin dynamics. (**A**) Measurement procedure. Left: Example kymograph of the bound ADF/Cofilin density as a function of space and time. Colored lines show three example paths in time and space along which the amount of bound ADF/Cofilin was measured. Inset: Schematic of the growing network at different time points, Right: The measurements of the bound ADF/Cofilin density along the paths shown on the left. Dotted lines show initial increase. (**B**) Normalized (to have mean one) values of the initial increase as a function of starting time for all networks. Colors represent different starting times, compare red, green and blue paths in the example network in *Figure 3A*. Inset: Predictions for local depletion and no local depletion of free ADF/Cofilin. (**C**) Simulation of (1 - 2). Snapshot at time t = 40 min showing the concentration of free ADF/Cofilin. Parameters: V = 1.16 μm/min, $r_B = 0.5$/min/μM, $r_U = 0.31$/min, A = 50 μM, $C_0 = 125$ nM. Inset left: Concentration of free ADF/Cofilin along the dotted line. Inset right: Zoom around the network. Colors represent concentration of free ADF/Cofilin. (**D**) Time snapshots of the same simulation showing the concentration of free ADF/Cofilin, colors as in *Figure 3C*. Below: Average amount of free ADF/Cofilin in the area covered by the network (black rectangle in *Figure 3D*).

DOI: https://doi.org/10.7554/eLife.42413.009

The following figure supplement is available for figure 3:

**Figure supplement 1.** Comparing simulated and measured amounts of bound ADF/Cofilin.

DOI: https://doi.org/10.7554/eLife.42413.010

respectively. Since the experimental chamber's depth in $z$-direction is much smaller than all characteristic dimensions in $x$- and $y$-directions, we use a 2D setting for modeling. In the simulations, an actin network of width $W$ and length $L(t) = V \times t$ is positioned in the middle of the experimental chamber. The model consists of the following equations:

$$\partial_t C_B = -V\partial_y C_B + r_B A C_F - r_U C_B, \tag{1}$$

$$\partial_t C_F = D\Delta C_F - r_B A C_F + r_U C_B. \tag{2}$$

Here *Equations (1) and (2)* describe the drift of bound and diffusion of free ADF/Cofilin molecules, respectively, and the reactions of ADF/Cofilin binding to and slow unbinding from actin filaments. Boundary and initial conditions, potential actin saturation effects and the numerical procedure for solving the model equations are discussed in Appendix 1.

The model does not describe actin disassembly, as we model the effect of ADF/Cofilin on actin filaments below in the next section. Thus, we either assume the actin density to be constant for rough estimates (the measurements show that the actin density changes relatively little along the network before plunging at the trailing edge, see *Figure 3—figure supplement 1A*), or equal to the measured function of the $y$-coordinate to compare with the data.

We can use *Equation (2)*, with constant actin filament density $A$, to estimate roughly the local concentration of free ADF/Cofilin near the actin network and the rate of ADF/Cofilin binding at the leading edge (details in Appendix 1):

$$C_F \approx \frac{C_0 D + WLC_B r_u}{D + AWLr_B}, \qquad \dot{C}_B \approx \frac{r_B A C_0 D}{D + AWLr_B}. \tag{3}$$

When the network grows, its length $L$ increases, and hence, as shown by these formulas, the local concentration of free ADF/Cofilin near the actin network decreases with time, and so does the rate of ADF/Cofilin binding at the leading edge, in agreement with the measurements (*Figure 3B*). This provides a demonstration of the local depletion of ADF/Cofilin due to the diffusion and binding to the network. Note that these calculations are but a rough order-of-magnitude estimate; to be more precise, we simulated the full 2D model (1) - (2) using parameters estimated from our data and taken from the literature (details in Appendix 1) and find a significant depletion effect near the network where the free ADF/Cofilin concentration drops by as much as 50% (*Figure 3D*).

To further test the model, we used the measurements of the actin density along the networks giving us functions $A(y)$ for tens of the networks of various densities at a certain time after the actin growth was initiated, and simulated (1) - (2) with these functions. This allowed direct comparison of measured and predicted ADF/Cofilin concentrations along the network. *Figure 3—figure supplement 1A–B* shows that the model recapitulates the distance-dependent concentrations and relative amounts of bound ADF/Cofilin very well.

## Equilibrium length of actin network as a function of biochemical and geometric parameters

### Actin network fragmentation at the trailing edge

When one observes the time lapse data of actin network dynamics at the trailing edge, it becomes apparent that the network does not disassemble continuously, but rather small, micron-size, pieces of the network break off (*Figure 4A*). Thus, the network disassembles by macroscopic fragmentation. To capture this dynamics, we followed the theory introduced in *Michalski and Carlsson (2010)*, *Michalski and Carlsson (2011)* and modeled the network as a 2D ensemble of edges connected by nodes. We emphasize that this representation is highly idealized, and the that the edges do not stand for individual filaments, but rather represent actin filaments arrays; similarly, nodes are not individual physical Arp2/3 complexes, but are abstracted crosslinking and/or branching points. We model the disassembling effect of ADF/Cofilin by removing the nodes with certain rate, $P$. Once a piece of the network becomes disconnected from the main body of the network due to this edge removal, we assume that this piece diffuses away and we delete it. *Figure 4C* illustrates how the model works.

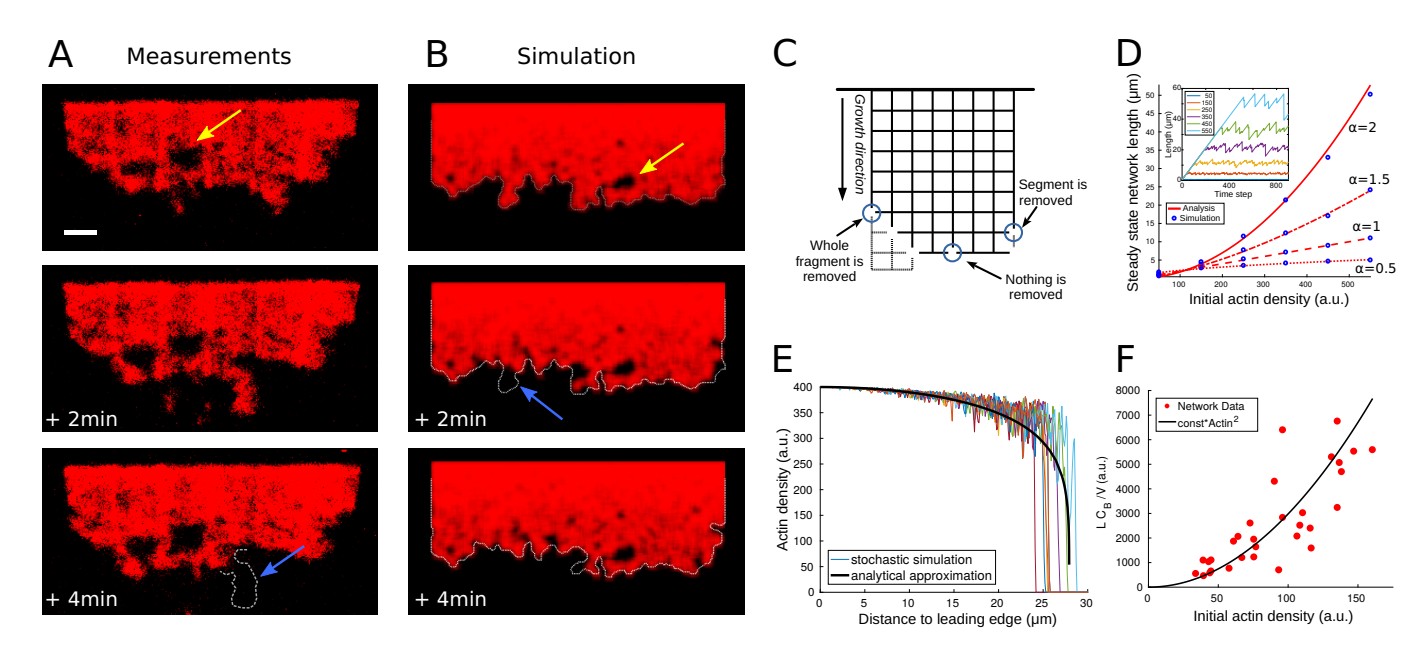

**Figure 4.** *Network fragmentation.* (**A**) Snapshots of experimental measurements of actin density (red) at three consecutive time points. The yellow arrow shows a hole in the network, the dotted outline and blue arrow the breakage of a large piece of network. The network width is 90 μm, the ADF/Cofilin concentration 200 nM, the bar is 10 μm. See *Figure 4—video 1*, right for the full time dynamics. (**B**) Simulation snapshots for three consecutive time steps. Colors and arrows as in *Figure 4A*, dotted lines show the shifted outline of the network in the previous time step. The same network width and speed and initial actin concentration as those measured for the network shown in *Figure 4A*, were used as model parameters in the simulation. See *Figure 4—video 1*, left for the full time dynamics. (**C**) Shown is the simulation setup. Effective nodes (branching and crosslinking points) are the vertices of the square lattice, while actin filaments are the edges of the square lattice. At every time step, the network is shifted in the growth direction. At each node, the breakage rate is a function of the local actin and global bound cofilin densities. The three circles show three different breakage events and their effects. (**D**) Comparison between the simulated equilibrium length and the analytical prediction using *Equation (4)* for different values of the initial actin density and exponent $\alpha$. Inset: The fluctuating network lengths as functions of time for various values of initial actin density for $\alpha = 2$. (**E**) Comparison between the actin density in the stochastic fragmentation simulation (thin lines) and the analytical approximation (thick black line). (**F**) Fit of the predicted quadratic dependence of the network equilibrium length on the actin density to the measurements of the equilibrium network length (L) normalized by the network speed (V) and the average concentration of bound cofilin ($C_B$). B, D, E: For details and parameters see Appendix 1.
DOI: https://doi.org/10.7554/eLife.42413.011

The following video is available for figure 4:

**Figure 4—video 1.** Video comparing fragmentation in the measurements (right) and the discrete network model (left) corresponding to *Figure 4A,B*.
DOI: https://doi.org/10.7554/eLife.42413.012

The key to the model behavior is setting rules that describe how the rate of breakage per node varies spatially. It is natural to assume that this rate is a function of local densities of filamentous actin and bound ADF/Cofilin. We also assume, for simplicity, that we can neglect a potentially complex effect of sequential biochemical reactions preceding the breakage events. Considering that both modeling and data shows that the bound ADF/Cofilin density changes little compared to actin near the trailing edge, we assume that the rate of breakage is a function of a spatially constant-bound ADF/Cofilin density. Thus, in the model, the rate of breakage (node disappearance) varies locally due to spatial variation of the local density of the actin network (we calculate the local density of the discrete network as a weighted average of the number of the network edges in the vicinity of a given node; details in Appendix 1). It is reasonable to assume that the node breakage rate would be a decreasing function of the actin filament density, as greater density of the actin filaments means also a greater density of the branching/crosslinking points, and effectively a number of such points per unit volume constitute a node.

Thus, we used the relation $P \propto \frac{C_B^\beta}{A^\alpha}$, where $C_B$ is the spatially constant concentration of bound ADF/Cofilin, $A$ is the local density of the discrete network, and $\alpha$ and $\beta$ are positive exponents that we varied in the simulations. We found that for many values of these exponents, the model was able to

recapitulate several key features of the observed actin network disassembly (*Figure 4B*, *Figure 4—video 1*). Specifically, the modeled dynamic networks were fragmenting at the trailing edge and forming holes near the edge. Analogously to the observations, we found that the modeled networks, after an initial period of growth, reached an equilibrium length, around which the network length fluctuated stochastically (*Figure 4D* inset). The model also predicted correctly the relatively small variation of the actin density along the network length, with a sharp drop at the trailing edge (*Figure 4E*).

For comparison with data, it is useful to derive an analytical approximation of the discrete, stochastic model. In Appendix 1, we introduce continuous deterministic densities of actin filaments and of broken nodes in the network, derive differential equations for these densities and solve these equations. This continuous deterministic model allows deriving analytical expression for the equilibrium network length $L$ as the function of three parameters, average bound ADF/Cofilin density, $C_B$, initial actin network density, $A_0$, and rate of the network growth at the leading edge, $V$:

$$L \propto V \frac{A_0^\alpha}{C_B^\beta}. \tag{4}$$

*Figure 4D* shows excellent agreement between the analytical approximation (4) and the corresponding network simulations.

To determine the values of the exponents $\alpha$ and $\beta$, we examined all networks in the experiments that have reached equilibrium, measured the values of parameters $L$, $V$, $A_0$ and $C_B$ (for $C_B$ we used the average across the network) for each network, and compared the actual equilibrium lengths to the ones predicted by *Equation (4)* based on the measured values of parameters $V$, $A_0$ and $C_B$. We found that for any $\alpha \in [1,3]$ and $\beta \in [0.5, 1.2]$, we had $R^2$-values of over 0.7, and $p<10^{-7}$. In the following we use $\beta = 1, \alpha = 2$ ($R^2 = 0.72$, $p<10^{-8}$). *Figure 4F* shows the quadratic dependence of the equilibrium network length on the initial actin density. This fit suggests that rate of disassembly of the effective network nodes is proportional to the bound ADF/Cofilin density and inversely proportional to the square of the local actin density. We discuss implications of this finding below.

## Balance between accumulation of ADF/Cofilin in longer networks and accumulation of network-breaking events predicts equilibrium network length

We can now combine the results from two models – for ADF/Cofilin binding and for network disassembly – to understand how the ADF/Cofilin dynamics and network fragmentation determine the equilibrium network length. In light of the relation

$$L \propto V \frac{A_0^2}{C_B}, \tag{5}$$

all that remains is to use the model from the previous section to estimate the average density of bound ADF/Cofilin $C_B$ and substitute the value into *Equation (5)*. In Appendix 1, we derived the following analytical estimate, based on the analysis of *Equations (3) and (1)*:

$$C_B \propto \frac{r_B A_0 C_0 L}{V} \times \frac{D}{r_B A_0 WL + D}, \tag{6}$$

which provides an explicit formula for the average density of bound ADF/Cofilin as a function of the leading edge actin density, rate of the network growth at the leading edge, the network dimensions and initial ADF/Cofilin concentration. This estimate has a simple interpretation: The first factor gives the average amount of the bound ADF/Cofilin in the absence of depletion. This amount is proportional to the actin density, initial ADF/Cofilin concentration and network length because the ADF/Cofilin binding rate is proportional to the actin density and the initial ADF/Cofilin concentration. The factor $L/V$ gives the characteristic time scale for ADF/Cofilin binding, that is longer/slower networks allow more time for ADF/Cofilin binding than shorter/faster networks. The second factor in *Equation (6)* represents a depletion factor, between 0 and 1, which shows by which fraction the local free ADF/Cofilin concentration near the network is decreased relative to the initial concentration $C_0$. The

larger the network (width $W$ or length $L$ or both are large), or the denser the network ($A_0$ is large), the more ADF/Cofilin is depleted. Finally, faster diffusion reduces the effect of depletion.

Note that the estimated amount of bound ADF/Cofilin in *Equation (6)* depends on the equilibrium length $L$ itself. Thus, the network equilibrium length is determined by the balance between two feedbacks (*Figure 5A*): the network length is shortened by higher ADF/Cofilin density, while the bound ADF/Cofilin density is increased by the network length. Mathematically, the first feedback is expressed by *Equation (5)* and effectively gives the bound ADF/Cofilin density as the decreasing function of the network length, while the second feedback is expressed by *Equation (6)* that gives the bound ADF/Cofilin density as the increasing function of the network length (*Figure 5A*). Together, these two equations constitute an algebraic system of equations for two variables – $L$ and $C_B$ – that has a unique solution for each value of four parameters, $A_0, C_0, V, W$, given graphically by the intersection of two curves for the relations $C_B(L)$ given by *Equation (6)* and *Equation (5)*, as shown in *Figure 5A*. In particular, since these two curves will always intersect, the network will reach some equilibrium length for any parameter combination. The effect of varying individual factors can now easily be understood (*Figure 5B–D*) and allows us to elucidate the experimental observations from *Figure 2*: Increasing the ADF/Cofilin concentration leads to more bound ADF/Cofilin and thereby shorter networks (*Figure 5B*). Increasing network density leads to less disassembly, and also to more depletion, and denser networks grow longer (*Figure 5D*). *Figure 2B* shows very good agreement between the model and the measurements. In the second experiment in *Figure 2C–D*, wider networks were less affected by ADF/Cofilin. The model suggests that this is because wider networks lead to more depletion and hence longer networks (*Figure 5C*), again in quantitative agreement with the measured lengths (*Figure 2D*).

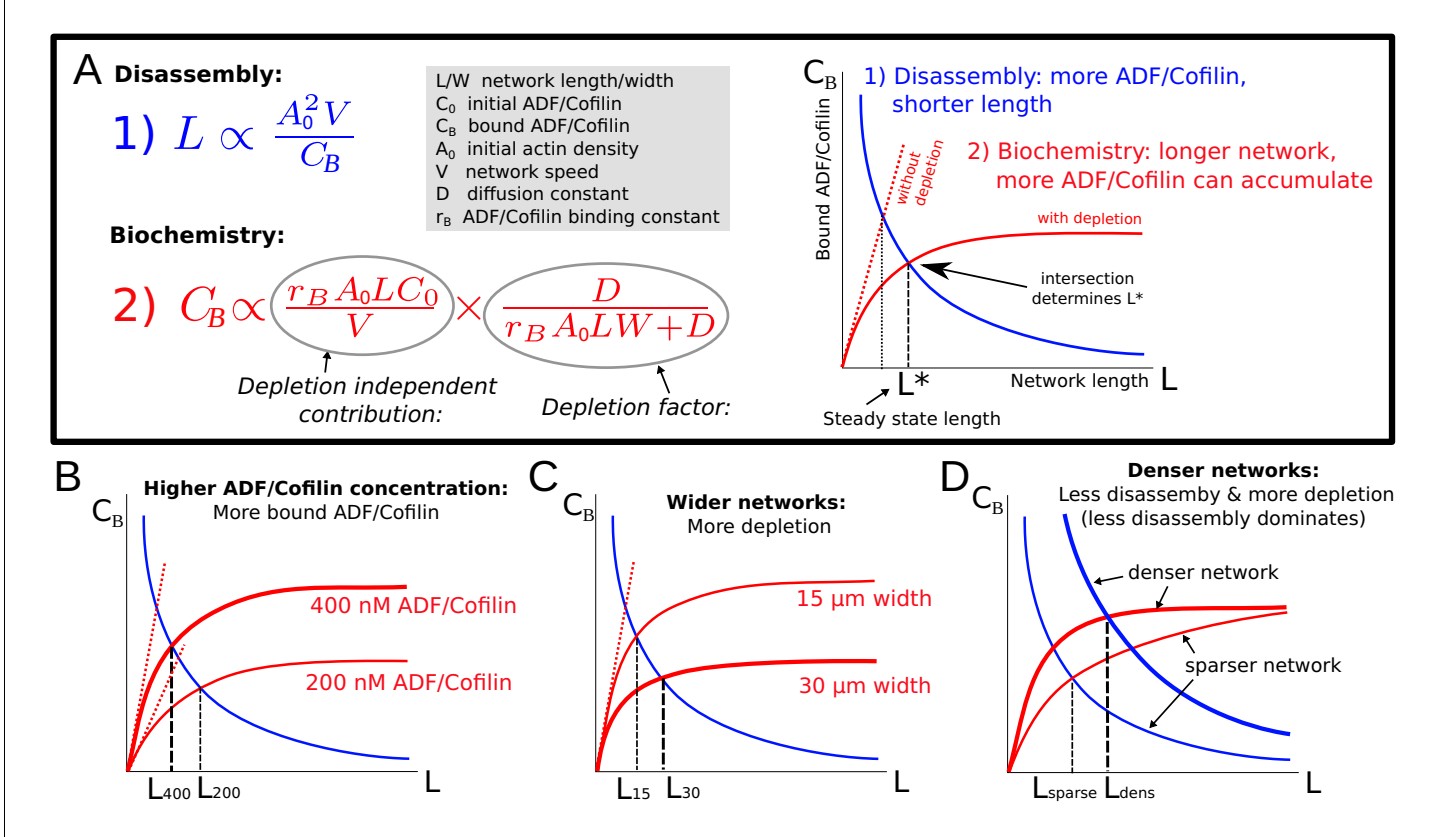

**Figure 5.** Model explanation. (A) Explanation of the two effects determining equilibrium network length: Feedback 1 (*Equation (5)*, blue equation and curve) shows the effect of the disassembly. Feedback 2 (*Equation (6)*, red equation and curve) shows the effect of ADF/Cofilin binding dynamics. Right: The intersection of the two curves in the $(L, C_B)$ plane determines the equilibrium network length marked by $L^*$. (B-D) Effect of varying the ADF/Cofilin concentration, the network width and the network density on the equilibrium network length.
DOI: https://doi.org/10.7554/eLife.42413.013

## ADF/Cofilin regulates steering of heterogeneous networks

In *Boujemaa-Paterski et al. (2017)* we found that network heterogeneity – varying actin filament density along the network leading edge – induces network steering, in the sense that the heterogeneous network grows curved. We explained this effect by the observation that the denser part of the network grows faster than the less dense part. Since these two parts of the network are interconnected, the only way for two network parts of different lengths to stay connected is if they grow along the arc of a circle. Then the faster part with the long axis further from the circle's center can grow longer, while advancing along the same arc length as the slower part (*Figure 6D*, left). This argument was purely geometric and implicitly assumed that the networks are plastic, bending freely. In fact, the networks are likely elastic or viscoelastic (*Gardel et al., 2004a*), which affects their bending behavior.

To simulate the steering heterogeneous network, we modeled the two networks as two elastic beams growing side-by-side. The networks had different densities and different growth speeds; we took the values of those parameters from the data (*Figure 6D*). We used the result from *Gardel et al. (2004a)* for random and isotropic actin network indicating that the network elasticity $E$ scales with actin density $A$ as $E \propto A^\tau, \tau \approx 2.5$. Note though that model predictions are not very sensitivecto the exact value of the exponent $\tau$, and that a few different values of parameter $\tau$ were

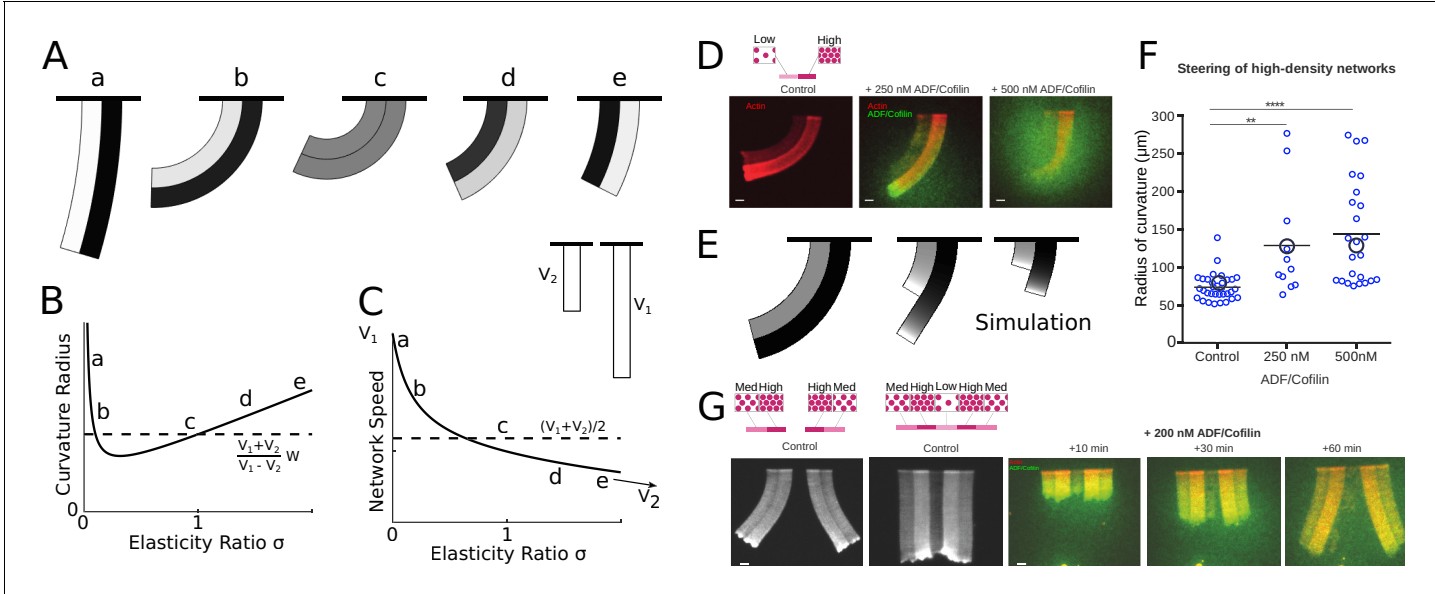

**Figure 6.** ADF/Cofilin controls the steering of heterogeneous actin network. (**A-C**) Modeling steering without ADF/Cofilin. (**A**) Model predictions of how differences in density influence the curvature and length of the heterogeneous LM. Darker colors signify denser networks, in all cases the left LM grows half as fast as the right LM. (**B,C**) Dependence of the curvature radius and speed of the heterogeneous LM on the elasticity ratio $\sigma$. The dashed line shows reference values, a-e mark the examples shown in *Figure 6A*. (**D-F**) Influence of ADF/Cofilin on the heterogeneous LM for concentrations 0, 250 nM and 500 nM. D. A pattern was generated with an array of spots of two distinct densities (left: low density, right: high density) both coated with the same concentration of NPFs. The heterogeneous pattern induces the growth of a heterogeneous actin network that steers toward the actin network with the lower density (left panel). Addition of 200 nM ADF/Cofilin selectively disassembles the low-density networks (middle panel). Addition of 500 nM ADF/Cofilin fully disassembles the low-density actin network and disassembles partially the high-density network (right panel). Snapshots were taken at 30 min after the addition of the actin monomers. (**E**) Simulated network shapes and network densities, darker colors signify denser networks. (**F**) Measured curvature radius, experimental (small blue circles) and simulated (large black circles). For the simulations we calculated an average curvature radius. (**G**) ADF/Cofilin induces steering within heterogeneous actin networks. We generated complex patterns made of heterogeneous spots density (medium and high spot densities) that are connected (right) or not (left) by a low-density pattern. Addition of 200 nM ADF/Cofilin selectively disassembles the low-density actin network induces the steering of the medium/high heterogeneous actin networks. See *Figure 6—video 1* for full time dynamics. (**A-C**), E: Details in Appendix 1.

DOI: https://doi.org/10.7554/eLife.42413.014

The following video is available for figure 6:

**Figure 6—video 1.** Selective disassembly of heterogeneous networks by ADF/Cofilin induces steering.
DOI: https://doi.org/10.7554/eLife.42413.015

reported, including $\tau \approx 0.5$ (*Bieling et al., 2016*) for branched actin networks, which are neither random nor isotropic (discussed in further detail in Appendix 1). We modeled the networks as two attached beams of width $W$, growing at speeds $V_1$ and $V_2$, with elastic moduli $E_1$ and $E_2$. In the absence of ADF/Cofilin we can assume that the densities and hence elasticities stay constant along the network. In Appendix 1 we demonstrated, that in mechanical equilibrium, the heterogeneous network forms a bent shape with constant radius of curvature $R$ (*Figure 6A–C*):

$$R = W\left(\frac{V_1 + V_2}{V_1 - V_2} + (\sigma - 1)\frac{V_1\sigma - V_2}{4\sigma(V_1 - V_2)}\right),\tag{7}$$

growing with speed:

$$V_h = \frac{V_1 + V_2}{2} - \frac{(V_1 - V_2)(V_1^2\sigma^2 - V_2^2)}{2(V_1^2\sigma^2 + 6V_1V_2\sigma + V_2^2)}\tag{8}$$

Here $\sigma = E_2/E_1$ is the ratio of the elastic moduli. The dependencies of the curvature and heterogeneous network speed on parameter $\sigma$ are depicted in *Figure 6B and C*, respectively. Note that if both networks have the same elastic properties, that is $\sigma = 1$, the heterogeneous network elongates with the average speed slightly less than $(V_1 + V_2)/2$, and the radius of curvature has the much simpler form $R = W(V_1 + V_2)/(V_1 - V_2)$, an approximation that has been used in *Boujemaa-Paterski et al. (2017)*. The steering direction (right or left) is solely determined by which part of the network grows faster – the heterogeneous network always steers towards the slower sub-network. Differences in elasticity, however, can influence the amount of steering in a complex way. *Figure 6A–C* shows that if one of the networks is very sparse (and hence weak elastically), the heterogeneous network becomes almost straight. There is a maximal steering curvature achieved for a certain elasticity ratio depending on the ratio of the speeds of the sub-network growth.

To asses the effect of ADF/Cofilin on heterogeneous networks, we used the model from the previous sections to calculate the equilibrium lengths of the two sub-networks and simulate the heterogeneous networks. Since the two sub-networks compete for the same pool of ADF/Cofilin, we need to adjust the depletion factor in *Equation (6)*. As described in Appendix 1 we can determine two equilibrium lengths $L_1$ and $L_2$ of the sub-networks. Effectively both networks will reach longer lengths together than in isolation, since there is more local depletion of ADF/Cofilin in the combined network. In addition, the sparser network is affected more by the depletion, as the denser networks 'uses up' disproportionately more ADF/Cofilin. Also, the network densities are not constant along the sub-networks, thereby leading to varying elasticities along the network. In terms of the model, this means that the parameter $\sigma$ becomes a function of the distance from the leading edge. Finally, the sparser sub-network has a trailing edge much closer to the leading edge than the dense one. Altogether, these factors mean that in the presence of ADF/Cofilin, the heterogeneous network will initially (closer to the leading edge) have the same curvature as without ADF/Cofilin. Further away the curvature decreases until the shorter sub-network fully disassembled, after which the longer sub-network is the only one remaining, and it continues to grow straight. *Figure 6E* shows that numerical simulations confirm these arguments and generate predictions for various ADF/Cofilin concentrations.

We imaged the curving heterogeneous networks (*Figure 6D*) and found that indeed increased ADF/Cofilin concentration straightens the combined network (*Figure 6F*) due to selective disassembly of the sparser sub-network and relieving the elastic constraint on the denser sub-network. The imaged network shapes appear qualitatively like the predicted shapes, and the measurements of the average curvatures give values similar to those predicted by the model (*Figure 6F*). Note, that the curvature changes very little on average when ADF/Cofilin concentration is increased from 250 to 500 nM because in both cases the sparser sub-network is almost completely disassembled.

An illustration that the effect of ADF/Cofilin can not only straighten, but also induce steering in heterogeneous networks, which grow straight in the absence of ADF/Cofilin, is given by the assay shown in *Figure 6G* and *Figure 6—video 1*. In this assay, the sparse sub-network was in the middle; two denser networks were at the sides of this central sub-network, and two more sparse sub-networks flanked the denser ones at the edges. Without ADF/Cofilin, such a combined network grew straight due to its mirror symmetry. Upon addition of ADF/Cofilin, the sparse sub-network in the middle was selectively disassembled, isolating the right and left heterogeneous networks from each other, which led to their steering away from each other.

# Discussion

## Summary of the results

We found that addition of ADF/Cofilin switched the actin networks' steady length increase to a 'global treadmilling' regime, in which the networks, after an initial growth stage, reach a dynamic equilibrium, with the network growing at the leading edge and falling apart at the trailing edge, and its length fluctuating around a constant. We observed that at the trailing edge, the network was stochastically fragmented into little pieces, rather than depolymerizing microscopically. Experiments showed that the equilibrium network length decreases with ADF/Cofilin concentration, and increases with the actin density and growing speed. The novel and counter-intuitive observation that the equilibrium network length increases with network width motivated the formulation of a computational model for ADF/Cofilin dynamics and subsequent comparison between simulated and measured spatio-temporal distributions of ADF/Cofilin and actin filament density. This led to a new insight: ADF/Cofilin is locally depleted from the solution by binding to actin filaments, which has a profound effect on actin disassembly, explaining why wider treadmilling networks are longer. While the effect of local depletion of actin monomers due to binding to actin filaments was recently reported both in vitro (*Boujemaa-Paterski et al., 2017*) and in vivo (*Dimchev et al., 2017*), the effect of local depletion of an actin accessory protein is reported here for the first time, to the best of our knowledge. This points to the possibility that similar depletion effects of other actin-binding proteins could be important for actin network dynamics.

We find that a single rate of disassembly, proportional to the local bound ADF/Cofilin density and inversely proportional to the square of local actin network density, can reproduce all experimental results. As a result, we were able to describe the dynamic equilibrium of actin networks with a simple formula enabling us to predict the length of the actin network as a function of its width, actin filament density, ADF/Cofilin concentration and growth rate. Finally, we made the observation that the radius of curvature of heterogeneous networks increases with the ADF/Cofilin concentration. A model suggests that ADF/Cofilin mediated disassembly effectively changes the elasticity of the networks in a spatially graded way, which affects the network curvature of heterogeneous growing networks. Thus, ADF/Cofilin can locally regulate the steering of heterogeneous networks.

## Relation to previous studies

Our observations and modeling results are in agreement with previous studies: ADF/Cofilin was observed to be distributed roughly uniformly across keratocyte's and fibroblast's lamellipodia, with a narrow ADF/Cofilin-free zone at the leading edge (*Svitkina and Borisy, 1999*). Similarly, in in vitro actin tails, the ADF/Cofilin density increased sub-linearly along the tail away from the leading edge, with the small ADF/Cofilin-free gap near that edge (*Reymann et al., 2011*). Just like our model, the theory in *Michalski and Carlsson (2010)* predicted an initial slow actin filament density decay followed by an abrupt decay at the edge of the tail. Such actin density behavior in lamellipodia of motile keratocytes was reported in *Barnhart et al., 2011*; *Raz-Ben Aroush et al. (2017)* and other experimental studies. The reason for this density behavior is the cooperative nature of network fragmentation, which accelerates non-linearly at low actin filament densities and leads to an abrupt falling apart of the network at the trailing edge (*Michalski and Carlsson, 2010*, *Michalski and Carlsson, 2011*).

Our model predicts that effective node-breaking events in the network take place on the scale of one per hundred seconds per micron. This is in agreement with measured severing times in vitro per micron of a filament of hundreds of seconds for 150 nM of ADF/Cofilin and tens of seconds for 1000 nM of ADF/Cofilin (*Chin et al., 2016*). The predicted proportionality of this rate to the ADF/Cofilin density is in agreement with the observation of the linear proportionality of the debranching to the ADF/Cofilin concentration at low concentrations (*Blanchoin et al., 2000*). Similar to *Michalski and Carlsson (2010)*, we found that, remarkably, the properties of the actin networks with actin subunits switching between many chemical and physical states can be described by a single effective disassembly rate, proportional to a certain mean of the chemical transition, severing and debranching rates. Note that the explanation for the abrupt disassembly at the trailing edge is not the abrupt increase of the rate of the node removal beyond a threshold of the cofilin decoration, but rather two-stage nature of the disassembly. First, node removal 'primes' the network for the disassembly;

second, the disassembly has an 'avalanche' character due to the positive feedback between the actin drop and node disappearance.

Just as *Michalski and Carlsson (2011)*, our model predicts that the network's width remains constant along the length, which we also observed. In vivo, this property of the lamellipodial networks is most clearly apparent in keratocytes' lamellipodial fragments (*Ofer et al., 2011*). Actin comet tails of intracellular pathogens also sometimes appear to have a constant width (*Akin and Mullins, 2008*), while under other conditions the tails taper as they decay (*Carlier et al., 1997*). We have to note that most of the previous measurements showed more gradual decrease of the actin density in the actin tails (*Lacayo et al., 2012*; *Cameron et al., 1999*; *Rosenblatt et al., 1997*). Yet, a few in vitro reconstitutions (*Loisel et al., 1999*; *Reymann et al., 2011*) revealed the abrupt actin density drop at the trailing edge. Also, abrupt actin density decrease at the rear of the lamellipodia in keratocyte cells and fragments (*Barnhart et al., 2011*; *Ofer et al., 2011*) was observed.

Directly, the 'macroscopic' fragmentation into micron-sized pieces was only observed in in vitro reconstitution experiments (*Reymann et al., 2011*). Indeed, it would be hard to imagine breakage of the microns-size fragments from a few micron-long lamellipodia. However, it is not out of question that such fragmentation could take place at the trailing edge of longer, $\approx 10\,\mu m$ long, lamellipodia. Imaging such a process is a challenging problem for the future. Note also that some of the images of the actin comet tails (for example, see *Figure 1* in *Cameron et al., 1999*) show patchy tail density at the trailing edge, which could be interpreted as the disassembly through fragmentation. Our results also are relevant to the 'microscopic' fragmentation through breaking actin filaments into small oligomers inferred from in vivo data in *Berro et al. (2010)*; *Raz-Ben Aroush et al. (2017)*.

Previous modeling showed that the length of the treadmilling network is (i) proportional to the polymerization velocity, (ii) is inversely proportional to the ADF/Cofilin density, (iii) and scales linearly with actin concentration (*Michalski and Carlsson, 2010*). Fitting our theoretical predictions to our data agrees with these previous predictions, with the exception that the network length is proportional to the square of the actin filament density at the leading edge. The reason for the difference, most likely, is that in *Michalski and Carlsson (2010)*, the network node breaking rate was proportional to the ADF/Cofilin density, and the node density scaled with the actin density. We suggest, similarly, that the network node (effectively, cross-linking and entanglement) density does scale with the actin density; however, the breaking (debranching and severing) rate is proportional to the ratio of the ADF/Cofilin to actin density, which is effectively the length density of ADF/Cofilin along actin filaments. Then, the ratio of the node density to the breaking rate per node, proportional to the square of the actin density divided by the ADF/Cofilin density, determines the network length.

Addition of ADF/Cofilin was shown to shorten *Listeria* actin tails (*Carlier et al., 1997*; *Rosenblatt et al., 1997*); proportionality of the *Listeria* actin tails' lengths to the polymerization rate at the leading edge was demonstrated in *Theriot et al. (1992)*, and proportionality of the lamellipodial length in motile keratocytes' fragments to the actin growth rate at the leading edge was reported in *Ofer et al. (2011)*. Interestingly, network length as a function of width was predicted to be linearly increasing and then saturating, but saturation happens when the width is on the order of 20 mesh sizes, on the micron scale (*Michalski and Carlsson, 2011*), and so this is unrelated to the effect that we report in this study.

## Novelty of our findings and relevance to in vivo networks

We established a simple formula that allows estimating the network length, $L$, as a function of a wide range of geometric and biochemical parameters: Actin filament density at the leading edge, $A_0$, speed of actin growth at the leading edge, $V$, width of the network, $W$, and initial ADF/Cofilin concentration, $C_0$ (*Figure 5*, *Figure 7*):

$$L = k_1 \frac{A_0^2 V}{C_B}, \quad C_B = k_2 \frac{r_B A_0 C_0 L}{V} \times \frac{1}{1 + \frac{r_B A_0 W L}{D}},$$

Here, $C_B$ is the density of ADF/Cofilin bound to the network, $D$ is the ADF/Cofilin diffusion coefficient in the solute, $r_B$ is the ADF/Cofilin binding coefficient, $k_1 \approx 1\,s/\mu M$ is a parameter determining the magnitude of the effective debranching and/or severing rate, and $k_2 \approx 1/2$ is a non-dimensional parameter.

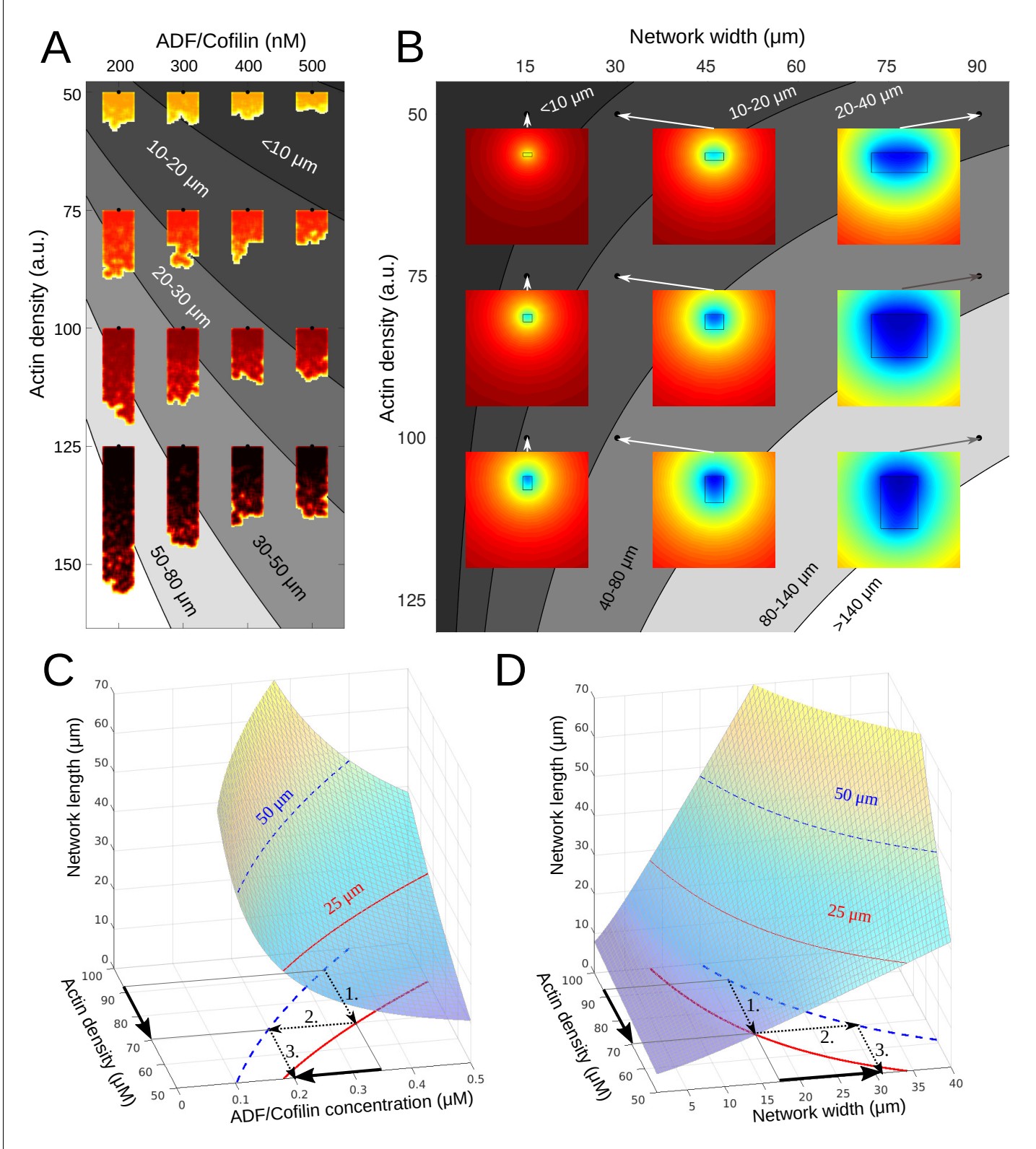

**Figure 7.** Phase diagram. (**A**) Depicted are the predicted equilibrium networks lengths, if ADF/Cofilin concentration and actin density are varied. Gray colors represent regions of similar equilibrium network length. Shadings of red show actin density (black = high, yellow = low). The network shapes were calculated using the fragmentation model. (**B**) Depicted are the predicted equilibrium networks lengths, if network width and actin density are varied. Gray colors in the background represent regions of similar equilibrium network length. Colored insets show the simulated amount of free ADF/

*Figure 7 continued on next page*

*Figure 7 continued*

Cofilin (blue = low, red = high). (**C-D**) Depiction of how different combinations of control variables can lead to the same network length. See text for details.

DOI: https://doi.org/10.7554/eLife.42413.016

One consequence of these results is that there are different ways for a cell to achieve the same network length. For example, if the actin density drops and hence the network becomes shorter, there are two ways to re-establish the original network length (*Figure 7A–D*): either the ADF/Cofilin concentration has to be decreased, or the network has to become wider. In both cases, the procedure is as follows (compare arrows in *Figure 7C–D*): 1. The drop in actin density leads to a new, shorter equilibrium length, 2. Through intersection with the constant-network length-level curves an alternative parameter combination can be identified, which gives the original network length. 3. This gives the new (lower) ADF/Cofilin concentration in solution or new (higher) network width necessary to maintain the original network length. This dynamic equilibrium underlies the network's ability to adapt to external changes. Indeed, cycles between low and high loads generate branched actin networks with different densities (*Bieling et al., 2016*). Our model explains how the system will respond to adjust its dimension according to these changes. Similarly, membrane tension affects the lamellipodium's actin filament density (*Mueller et al., 2017*). The dynamic equilibrium model predicts how tension sensing will be counterbalanced to preserve the dimension of the growing network.

We found that diffusion of ADF/Cofilin in the solution and binding to the growing actin network can locally deplete the cytoplasmic ADF/Cofilin, which makes wider and denser actin networks grow longer (*Figure 7B*). Quantitatively, whether the depletion is significant or not, is determined by the magnitude of the non-dimensional quantity $\frac{r_B A_0 W L}{D}$: if this factor is smaller than 1 (e.g. when the network width and length are small enough), there is no significant depletion; otherwise, there is.

So, is the ADF/Cofilin depletion relevant in vivo? The following estimates give a clear positive answer. Using $r_B \approx 0.01/(s\ \mu M)$ (*Tania et al., 2013*), we estimated that $r_B A_0 \approx 1 - 10/s$ (F-actin density is in the range of hundreds of $\mu M$ for observed branched networks (*Koestler et al., 2009*; *Urban et al., 2010*), and $D \approx 10\ \mu m^2/s$ (*Tania et al., 2013*). Thus, for actin tails propelling intracellular pathogens and organelles, for which $W \approx 1\ \mu m$ and $L \approx 3\ \mu m$, we have: $\frac{r_B A_0 W L}{D} \approx 0.3 - 3$, and the depletion of ADF/Cofilin is moderate but present. This effect is even more pronounced for the branched networks in cells. For example, the characteristic dimensions of the lamellipodial network in fish keratocyte cell is $W \approx L \approx 10 m$, so $\frac{r_B A_0 W L}{D} \approx 10 - 100$, and there is very significant depletion effect for ADF/Cofilin. Even for shorter lamellipodia in many other cells, with $L \approx 1 - 4 m$, $W \approx 10 m$, $\frac{r_B A_0 W L}{D} \approx 1 - 40$, and the depletion effect is not negligible.

In the limit when the depletion is in effect, we predict that $C_B \approx k_2 \frac{D C_0}{V W}$, and the length of the actin networks can be estimated by the simple formula:

$$L \approx \frac{k_1}{k_2} \frac{A_0^2 V^2 W}{D C_0}.$$

The following parameter values then allow to estimate the length: For the rapidly growing actin comet tail and lamellipodial networks, $V \approx 0.1\ \mu m/s$ (*Theriot et al., 1992*; *Barnhart et al., 2011*). The ADF/Cofilin concentration in many animal cells is on the order of tens of $\mu M$ (*Pollard et al., 2000*). Then, for the actin comet tail (for which $W \approx 1\ \mu m$), we predict the length $L \approx 10 - 20\ \mu m$. Note that in two previous in vitro reconstitution studies (*Loisel et al., 1999*; *Reymann et al., 2011*), the observed tail length, when ADF/Cofilin was the only depolymerization factor, was on the order of 20 $\mu m$, in line with our estimate. For the actin tails observed in cells and in cell extracts, the length is a few-fold lower – on the order of a few microns – which is in agreement with a few-fold disassembly acceleration effect generated by molecular cofactors of ADF/Cofilin (see discussion below). Similarly, for the lamellipodial networks (for which $W \approx 10\ \mu m$), we predict the length $L \approx 100\ \mu m$, which is an order of magnitude longer than observed. This is a clear indication that the action of ADF/Cofilin molecular cofactors, in addition to possible nonlinear scaling of the ADF/Cofilin concentration effect must be in effect.

The clear in vivo relevance of the branched network steering is illustrated by recent observations that flat Arp2/3-governed sheets of branched actin regulate pathfinding of cells in 3D ECM (*Fritz-Laylin et al., 2017*). The question of how motile cells turn is attracting growing attention. A number of turning mechanisms were elucidated. As expected, chemotaxis-related biochemical pathways upstream of the actin network mechanics can regulate lamellipodial steering (*Yang et al., 2016*). However, mechanics, architecture and turnover of the network at the leading edge can lead to steering even in the absence of the upstream control. Examples of such mechanisms include Rac-Arpin nonlinear feedbacks regulating of the Arp2/3-branching activity (*Dang et al., 2013*), spatially graded thymosin β4 mediated control of the lamellipodial turning (*Roy et al., 2001*) and monomer-diffusion mediated steering of heterogeneous actin networks (*Boujemaa-Paterski et al., 2017*). Steering of intracellular pathogens by curving their actin tails depends on harnessing viscoelastic deformations of the actin tails and polymerization forces on the curved pathogen surface to generate actin growth asymmetries (*Lacayo et al., 2012*). Motile cell turning can also rely on alternating types of actin networks (*Diz-Muñoz et al., 2016*) and on crosstalk between actin and microtubule dynamics (*Buck and Zheng, 2002*). Lastly, cells also can steer from the rear of the networks, by actin-myosin contraction asymmetry mechanism (*Nickaeen et al., 2017*). Our findings add important additional control mechanism of tuning curvatures of the heterogeneous networks by ADF/Cofilin-mediated changes to network elasticity.

## Model limitations and outstanding questions

Our experiments and modeling do not address the microscopic mechanism for the biological function of ADF/Cofilin, which is still debated. Our model is not explicitly microscopic and does not distinguish between ADF/Cofilin-mediated severing, acceleration of disassembly at filament ends and debranching (*Chan et al., 2009*) (reviewed in *Blanchoin et al., 2014*). Similarly, the model took into account neither ATP hydrolysis on actin subunits and preferential binding of ADF/Cofilin to ADP-actin (*Blanchoin and Pollard, 1999*), nor cooperativity of ADF/Cofilin binding (*Hayakawa et al., 2014*), nor ADF/Cofilin-induced structural change and destabilization of filaments (*Pfaendtner et al., 2010*; *Suarez et al., 2011*; *Wioland et al., 2017*). Due to technical limitations, we did not explore very high ADF/Cofilin concentrations, at which over-decoration by ADF/Cofilin can lead to filament stabilization (*Andrianantoandro and Pollard, 2006*), and rate of debranching can become a nonlinear function of ADF/Cofilin concentration (*Chan et al., 2009*). Thus, we observed neither non-monotonic dependence of the severing activity on ADF/Cofilin concentration (*Andrianantoandro and Pollard, 2006*; *Pavlov et al., 2007*), nor independence of the lengths of *Listeria* actin comet tails on high ADF/Cofilin concentrations (*Rosenblatt et al., 1997*). Even though our model did not account for all this microscopic complexity, the model predictions are remarkably efficient, pointing out two important factors: Hydrolysis is fast enough so that only a micron- or sub-micron-size region near the very leading edge is affected by the hydrolysis state of the actin network, which is negligible when we deal with networks longer than a few microns. Also, as we note above, on the more macroscopic scale of the whole network, the microscopic complexity can be effectively combined into one overall disassembly rate.

We also did not address the emerging molecular complexity of the disassembly process: in vivo, ADF/Cofilin often acts in synergy with the ADF cofactor actin-interacting protein 1 (AIP1), twinfilin, coronin and Srv2/adenylyl cyclase-associated protein (*Kueh et al., 2008*; *Johnston et al., 2015*). One of the obvious effects of the disassembly cofactors is acceleration of the disassembly process. Concerted action of Twinfilin, Coronin and Aip1, when added to cofilin, was demonstrated to accelerate the disassembly by a few-fold, up to an order of magnitude (*Johnston et al., 2015*; *Chin et al., 2016*). This would bring down the estimates of the actin networks' lengths above to the observed values. One example of the extremely fast disassembly is actin patches in yeast, which are so small (micron scale) and have such a rapid dynamics (on the order of seconds) (*Berro et al., 2010*) that fast microscopic mechanisms employing additional molecular machinery, not accounted for in our study are likely involved. How the disassembly cofactors change the fragmentation scenario is also a great question for the future. One possibility is that these cofactors make the disassembly smoother: it was demonstrated that concerted action of Cofilin, Aip1 and Coronin first breaks filaments into small fragments, and then disassembles the fragments into monomers at such speed that effectively the disassembly is continuous (*Johnston et al., 2015*).

In addition, there are ADF/Cofilin-independent disassembly mechanisms, that is myosin-powered grinding of the actin network at the cell rear (*Wilson et al., 2010*). This synergy, added to complex nonlinear feedbacks between the branching, assembly and disassembly processes (*Tania et al., 2013*) and complex transport and partitioning of actin monomers and filaments in the cell (*Vitriol et al., 2015*; *Raz-Ben Aroush et al., 2017*) cause ADF/Cofilin to affect not only the disassembly, but also polymerization rate and network density. For example, higher ADF/Cofilin concentration can accelerate growth speed (*Aizawa et al., 1996*; *Carlier et al., 1997*). In the future, the in vitro and in silico studies will have to address these systems-level actin network dynamics. Last, but not least, cell actin networks integrate architectures other than Arp2/3-controlled branched lamellipodia and comet tails, and there is a delicate, incompletely understood dynamic balance between branched, bundled and other networks (*Blanchoin et al., 2014*). Dependence of the disassembly on network architecture was recently discovered (*Gressin et al., 2015*). Future models and experiments will have to investigate quantitative rules of the integrated global actin network dynamics.

## Conclusion

Our study leads to the important general conclusion that the cell is able to control the dynamic actin network length by adjusting either geometric, structural, or biochemical parameters, as needed. For example, if the network's width is dictated by the environment around the cell, then network's length can be regulated by tuning ADF/Cofilin concentration (*Figure 7A–D*). On the other hand, if the ADF/Cofilin concentration has to be tuned for timely disassembly of other actin structures, then the branched network's density or width can be changed in order to achieve necessary length (*Figure 7A–D*). In other words, there are multiple ways to set the dynamic balance of the biochemical and transport pathways regulating the global actin treadmill. This gives the cell sufficient flexibility in the control of the cytoskeletal geometry, without compromising requirements for mechanical and biochemical parameters to control multiple cytoskeletal functions.

## Materials and methods

**Key resources table**

| Reagent type (species) or resource | Designation | Source | Identifiers | Additional information |
|---|---|---|---|---|
| Biol. sample (Bovine) | Bovine Thymus | Slaughterhouse, SAINT EGREVE | | |
| Biol. sample (Rabbit) | Rabbit Muscle Acetone Powder | Pel-Freez Biologicals | Cat# 41995–2 | |
| Strain, strain background (E coli) | BL21(DE3) p Lys S | Merck | Cat# 69451 | |
| Strain, strain background (E coli) | Rosettas 2 (DE3) p Lys S | Merck | Cat# 71403 | |
| Peptide, recomb. protein | Mouse Capping proteins | Uniprot | $\alpha$ & $\beta$ subunits, P47754 and P47757 | |
| Peptide, recomb. protein | Human Profilin 1 | Uniprot | P07737 | |
| Peptide, recomb. protein | Yeast cofilin | Uniprot | Q03048 | |
| Peptide, recomb. protein | Human WASp pWA | Uniprot | P42768 | seq. 150–502 aa |
| Chem. compound, drug | mPEG-Silane, MW 30 k | Creative PEGWorks | Cat# PSB-2014 | |
| Chem. compound, drug | Alexa Fluor 488 C5 Maleimide | ThermoFisher Scientific | Cat# A10254 | |
| Chem. compound, drug | Alexa Fluor 568 NHS Ester | ThermoFisher Scientific | Cat# A20003 | |

*Continued on next page*

*Continued*

| Reagent type (species) or resource | Designation | Source | Identifiers | Additional information |
|---|---|---|---|---|
| Commercial assay or kit | Glutathione Sepharose 4B | GE Healthcare Life Sciences | Cat# 17075605 | |
| Commercial assay or kit | Ni Sepharose High Performance | GE Healthcare Life Sciences | Cat# 17526802 | |
| Software, algorithm | Matlab code for a standard numerical algorithm to solve the reaction-diffusion equations | This paper; **Source code 1** | | |

## Protein production and labeling

Actin was purified from rabbit skeletal-muscle acetone powder (*Spudich and Watt, 1971*). Actin was labeled on lysines with Alexa-568 (*Isambert et al., 1995*). Labeling was done on lysines by incubating actin filaments with Alexa-568 succimidyl ester (Molecular Probes). All experiments were carried out with 5% labeled actin. The Arp2/3 complex was purified from bovine thymus (*Egile et al., 1999*). Human WASp-pVCA (GST-WASp-pVCA) is expressed in Rosettas 2 (DE3) pLysS and purified according to *Boujemaa-Paterski et al. (2017)*. Human profilin is expressed in BL21 DE3 pLys S Echerichia coli cells and purified according to *Almo et al. (1994)*. Mouse capping protein is purified according to *Falck et al. (2004)*.

## Laser patterning

$20 \times 20$ mm$^2$ coverslips and cover glasses (Agar Scientific) were extensively cleaned, oxidized with oxygen plasma (3 mn at 30 W, Harrick Plasma, Ithaca, NY) and incubated with 1 mg ml$^{-1}$ of Silane-PEG overnight. Patterns of the desired density and area were printed on Silane-PEG-coated surfaces using a pulsed, passively Q-switched laser (STV-E, TeamPhotonics) that delivers 300 ps pulses at 355 nm. The laser power is controlled with a polarizer (iLasPulse device, Roper Scientific). Following laser patterning, patterned coverslips were coated with a solution of NPF at a concentration of 500 to 1000 nM for 15 min. The excess of NPFs was washed out with G-buffer (5 mM Tris-HCl [pH 8.0], 0.2 mM ATP, 0.1 mM $CaCl_2$ and 0.5 mM dithiothreitol (DTT)), and the surface was carefully dried.

## Reconstituted LMs

Assembly of reconstituted LMs was initiated in polymerization chambers of $20 \times 20$ mm$^2$ x 4.5 μm height by addition of the actin polymerization mix contained 6 μM actin monomers (containing 3% Alexa568-labeled actin), 18 μM profilin, 120 nM Arp2/3, 25 nM CP, in X buffer (10 mM HEPES [pH 7], 0.1 M KCl, 1 mM $MgCl_2$, 1 mM ATP, and 0.1 mM $CaCl_2$) and was supplemented with 1% BSA, 0.2% methylcellulose, 3 mM DTT, 0.13 mM 1,4-diazabicyclo[2.2.2]octane (DABCO), 1.8 mM ATP (*Boujemaa-Paterski et al., 2017*). When needed, the polymerization mix also included yeast cofilin purified according to *Suarez et al. (2011)* at a concentration of 125, 250, or 500 nM. We normalized the actin network fluorescence between assays using 0.2 μm TetraSpeck fluorescent beads (Molecular Probes).

## Image acquisition

Image acquisition was performed using an upright Axioimager M2 Zeiss microscope equipped with an EC Plan-Neofluar dry objective (x20, NA 0.75), a computer controlled fluorescence microscope light source X-Cite 120PC Q (Lumen Dynamics), a motorized XY stage (Marzhauser) and an ORCA-ER camera (Hamamatsu). The station was driven by MetaMorph software (Universal Imaging Corporation). The growth rates were calculated using ImageJ software.

## Mathematical modeling

Details about the mathematical modeling, analysis and simulation can be found in Appendix 1.

## Acknowledgements

This work was supported by grants from European Research council (741773 (AAA)) awarded to LB, Agence Nationale de la recherche (MaxForce, ANR-14-CE11) awarded to LB and MT. AI was supported by a fellowship from Université de Grenoble Alpes. AM and AM are supported by US Army Research Office grant W911NF-17-1-0417.

## Additional information

### Funding

| Funder | Grant reference number | Author |
|---|---|---|
| H2020 European Research Council | 741773 | Laurent Blanchoin |
| Agence Nationale de la Recherche | ANR-14-CE11 | Manuel Thery |
| Université Grenoble Alpes | Fellowship | Téa Aleksandra Icheva |
| Army Research Office | W911NF-17-1-0417 | Alex Mogilner |

The funders had no role in study design, data collection and interpretation, or the decision to submit the work for publication.

### Author contributions

Angelika Manhart, Data curation, Formal analysis, Investigation, Methodology, Writing—original draft, Writing—review and editing; Téa Aleksandra Icheva, Christophe Guerin, Tobbias Klar, Rajaa Boujemaa-Paterski, Data aquisition, Formal analysis; Manuel Thery, Conceptualization, supervision; Laurent Blanchoin, Alex Mogilner, Conceptualization, Supervision, Writing—original draft, Writing—review and editing

### Author ORCIDs

Téa Aleksandra Icheva (iD) https://orcid.org/0000-0002-4737-1509
Manuel Thery (iD) https://orcid.org/0000-0002-9968-1779
Alex Mogilner (iD) https://orcid.org/0000-0002-9310-3812

### Decision letter and Author response

Decision letter https://doi.org/10.7554/eLife.42413.025
Author response https://doi.org/10.7554/eLife.42413.026

## Additional files

### Supplementary files

• Source code 1. Source code file.
DOI: https://doi.org/10.7554/eLife.42413.017

• Transparent reporting form
DOI: https://doi.org/10.7554/eLife.42413.018

### Data availability

All data generated or analysed during this study are included in the manuscript and supporting files.

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

## Appendix 1

DOI: https://doi.org/10.7554/eLife.42413.019

### Effective diffusion coefficient of ADF/Cofilin in the branched actin network

Here, we test the hypothesis whether wider networks are less affected by ADF/Cofilin because dense actin networks hinder diffusion, and this does not allow enough time for ADF/Cofilin to access the middle of a network. To do so, we used the theory developed in **Novak et al. (2009)**, which examines how the presence of obstacles affects the effective diffusion constant of a particle. According to **Novak et al. (2009)**, the ratio between the ADF/Cofilin diffusion constant in the cytosol, $D$, and the effective diffusion constant of ADF/Cofilin in the branched actin network, $D_{\text{eff}}$, can be estimated as:

$$\frac{D_{\text{eff}}}{D} = \frac{(1 - \phi/\phi_c)^\mu}{1 - \phi}.$$

In the theory derived in **Novak et al. (2009)**, the actin network is represented as a collection of long cylindrical obstacles/filaments, through which ADF/Cofilin molecules diffuse. This theory estimates the ratio of the diffusion coefficients as the function of three parameters: $\phi$, $\phi_c$ and $\mu$. Exponent $\mu = 1.58$ was estimated in **Novak et al. (2009)** based on characteristic dimensions of the cylindrical obstacles. Parameter $\phi$ reflects the effect of the actin volume fraction on diminishing the diffusion coefficient and is given as $\phi = 1 - \exp(-V)$, where parameter $V$ depends on the sum of volumes of individual obstacles per unit volume, and is determined by the number of filaments per unit area, $f$, the average orientation angle of the filaments, $\alpha = 35°$, and the radii of actin filaments, $r_A = 3.5$ nm, and of ADF/Cofilin molecules, $r_C = 1.58$ nm, as follows:

$$V = \frac{f(r_A + r_C)^2 \pi}{\sin \alpha}. \tag{9}$$

Finally, parameter $\phi_c = 0.942$ characterizes the critically dense network, which completely obstructs the diffusion. A conservative estimate can be made by assuming a dense actin network with $f = 300/\mu\text{m}^2$. This gives estimates of $V = 0.042$ and $D_{\text{eff}} = 0.97D$. Thus, the effect of even a dense actin network on the ADF/Cofilin diffusion coefficient is but a few per cent and can be neglected.

### Determining initial ADF/Cofilin binding rate

To determine the ADF/Cofilin binding rate (see Sec. Spatio-temporal ADF/Cofilin dynamics and its local depletion in the main text), we used the experimentally measured concentrations of ADF/Cofilin and actin. We focused on the changes of bound ADF/Cofilin concentration at the beginning the network growth, since in this early stage we can neglect both ADF/Cofilin unbinding and depletion of free ADF/Cofilin. This means that we can assume that:

$$\partial_t C_B + V \partial_y C_B \approx b(A, C_0),$$

where $b(A, C_0)$ is the binding rate we would like to determine. Since we know the network growth speed, we can measure the increase of bound ADF/Cofilin $\dot{C}_B$ in moving patches of actin, that is we can directly measure $b(A, C_0)$. First, we examined networks with similar actin densities $A$, and found a strong correlation ($R = 0.69$, $p{<}10^{-3}$) between the binding rate $\dot{C}_B$ and initial concentration of ADF/Cofilin $C_0$ (**Appendix 1—figure 1A**). Next, we examined networks in the experiments with similar initial ADF/Cofilin concentrations $C_0$, but with varying actin densities $A$, and found a strong correlation between $\dot{C}_B$ and the actin density $A$ ($R = 0.69$, $p{<}10^{-5}$) (**Appendix 1—figure 1B**). Finally, we examined networks with varying values of parameters $C_0$ and $A$, and found that indeed the binding rate $\dot{C}_B \propto AC_0$ ($R = 0.51$, $p{<}10^{-5}$,

*Appendix 1—figure 1C*), justifying the use of the proposed mathematical form for the binding rate $b(A, C_0) = r_B A C_0$ at the beginning of the network growth. In the main text, we show, by comparison with the data, that in fact the form $b(A, C_0) = r_B A C_F$, that is the rate of binding being limited by the local, not initial, concentration of free ADF/Cofilin, $C_F$, leads to the model predictions that fit the data very well.

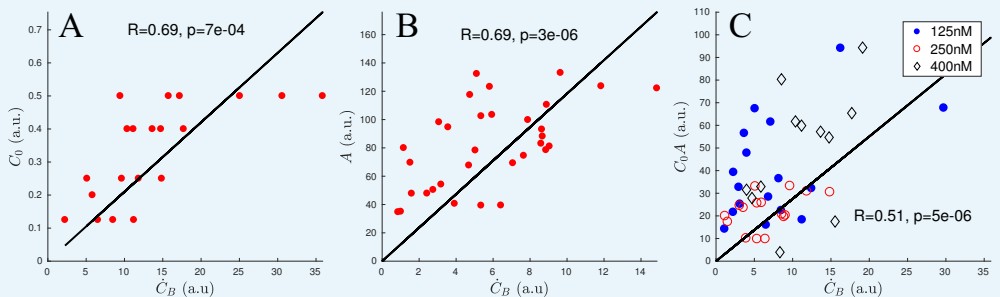

**Appendix 1—figure 1.** Scaling of ADF/Cofilin binding rate. (**A**) Correlation of $\dot{C}_B$ and $C_0$ using only networks with an actin density between 120 and 180 (a.u.). (**B**) Correlation of $\dot{C}_B$ and $A$ using only networks with an initial ADF/Cofilin concentration between 200 and 250 nM. (**C**) Correlation of $\dot{C}_B$ and $C_0 A$ using networks of varying initial ADF/Cofilin concentration and actin density.
DOI: https://doi.org/10.7554/eLife.42413.020

## Spatio-temporal ADF/Cofilin model: details and simulation

### Model

In this section, we provide details for the model of the ADF/Cofilin dynamics. All model parameters are gathered in *Appendix 1—table 1* below. We simulate the model in 2D with $(x, y) \in [-B/2, B/2] \times [-B/2, B/2] \subset \mathbb{R}^2$, where $B > 0$ is the size of the square-shaped domain. The density of free ADF/Cofilin molecules diffusing in the solute is denoted by $C_F(x, y, t)$, those bound to the actin network by $C_B(x, y, t)$. In the simulations, an actin network of width $W$ and length $L(t) = V \times t$ is positioned at $\mathcal{N} = [-W/2, W/2] \times [0, L(t)]$. The model consists of the following equations:

$$\partial_t C_B = -V \partial_y C_B + r_B A\, C_F - r_U C_B, \tag{10}$$

$$\partial_t C_F = D \Delta C_F - r_B A\, C_F + r_U C_B, \tag{11}$$

with the initial conditions $C_F(x, y, 0) = C_0$, $C_B(x, y, 0) = 0$, and the boundary conditions:

$$C_F(x, y, t) = C_0 \quad x = \pm B/2 \quad \text{or} \quad y = \pm B/2$$

$$C_B(x, 0, t) = 0.$$

**Appendix 1—table 1.** Simulation parameters of ADF/Cofilin binding/unbinding model.

**Variables & parameters**

| Name | Meaning | Value | Comment |
|------|---------|-------|---------|
| $C_F$ | diffusing ADF/Cofilin | in μM | simulated |
| $C_B$ | bound ADF/Cofilin | in μM | simulated |
| $V$ | network growth speed | ≈1-2 μm/min | measured |

*Appendix 1—table 1 continued on next page*

*Appendix 1—table 1 continued*

**Variables & parameters**

| Name | Meaning | Value | Comment |
|------|---------|-------|---------|
| $D$ | ADF/Cofilin diffusion constant | 600 µm²/min | from *Tania et al. (2013)* |
| $r_B$ | ADF/Cofilin binding rate | 0.5/min/µM | from *Reymann et al. (2011)* |
| $r_U$ | ADF/Cofilin unbinding rate | 0.31/min | from *Reymann et al. (2011)* |
| $A$ | actin density | 25-400 µM | estimated in *Boujemaa-Paterski et al. (2017)* |
| $C_0$ | initial ADF/Cofilin concentration | 125-500 nM | experimental set-up |
| $W$ | network width | 15-90 µm | experimental set-up |
| $B$ | domain length | 1 mm | reflects experimental set-up |
| $\Delta t$ | time step for transport operator | 1.5 min | |

DOI: https://doi.org/10.7554/eLife.42413.021

In the experiment, networks grow in a large, several square millimeter sized chambers, so that the total amount of ADF/Cofilin is not limiting, and moreover, over the time of the experiment, about 60 min, diffusion is not fast enough to diminish the ADF/Cofilin concentration in the solute farther than a few hundred microns from the growing network (for relevant estimates, see *Boujemaa-Paterski et al. (2017)*. For this reason, we performed the simulations in the area one millimeter in size, smaller that the size of the whole experimental chamber, but large enough so that the concentration of free ADF/Cofilin at its boundary is almost identical to the initial ADF/Cofilin solute concentration. This justifies using Dirichlet boundary conditions also for $C_F$, rather than no flux boundary conditions. For the actin density, we assume $A(x, y, t) \equiv 0$ whenever $(x, y) \notin N$. Within the network, we use two scenarios: The actin density is constant, or the actin density is a function of $y$ only, that is $A(x, y, t) = A(y)$, where we use the measured actin density along the network, averaged over its width and fitted using a smoothing spline. The smoothing avoids potential numerical problems when solving partial differential equations due to the roughness of the measured data. Finally, note that the macroscopic model does not account for an ATP-F-actin band, for which ADF/Cofilin has a much lower affinity. The reason is that such an ATP-F-actin band is very narrow (see Discussion for details).

We can roughly estimate the free ADF/Cofilin near the network and the rate of ADF/Cofilin binding to the network near the leading edge as follows. The flux of the free ADF/Cofilin to the network by diffusion, $(C_0 - C_F)D$, has to balance the "consumption" of the free ADF/Cofilin by binding to the network, that is

$$(C_0 - C_F)D \approx WL(r_B C_F A - r_U C_B),$$

which leads to the first formula in *Equation (3)*. The unbinding of ADF/Cofilin is very slow; besides, near the leading edge, $C_B \approx 0$, and so we arrive at the second formula in *Equation (3)*.

We note that ADF/Cofilin unbinding predicted by the model is slow; omitting the effect of ADF/Cofilin unbinding (i.e. setting $r_U = 0$ in the model) makes the fits to the data less perfect, however, the estimates of the overall amount of bound ADF/Cofilin changes little.

## Actin saturation

In *Equations (10)-(11)* we assume that the binding rate is proportional to the actin density, irrespective of how much ADF/Cofilin is already bound within the network. A more accurate model would involve defining $A_F = A - A_D$, where $A_D$ is the concentration of actin decorated with ADF/Cofilin, which is no longer available for further ADF/Cofilin molecules to bind, and $A_F$ is the F-actin density free of ADF/Cofilin. The new set of equations would then read

$$\partial_t C_B + V \partial_y C_B = r_B (A - A_D) C_F - r_U C_B,$$

$$\partial_t C_F = D \Delta C_F - r_B (A - A_D) C_F + r_U C_B,$$

$$\partial_t A_D + V \partial_y A_D = r_B (A - A_D) C_F - r_U C_B$$

In the equations we used the known stoichiometry of one ADF/Cofilin molecule binding to one actin subunit (**Kuhn and Bamburg, 2008**). Furthermore, decorated actin follows the same transport dynamics and has the same boundary conditions as bound cofilin. Hence, we can conclude that in fact $A_D = C_B$. If we now look at the relative order of magnitudes of the three reaction terms (stripping away the physical-chemical dimensions), we find that $r_B \times A \times C_F \approx 0.5 \times 100 \times 0.1 = 5$, $r_B \times C_B \times C_F \approx 0.5 \times 10 \times 0.1 = 0.5$ and $r_U \times C_B \approx 0.3 \times 10 = 3$. Here we use the estimate $C_B \approx 10 \, \mu M$ as shown in **Figure 3—figure supplement 1A**. We see that the term $r_B \times C_B \times C_F$, which is responsible for the correction of term $A$ by term $A - A_D$ is an order of magnitude smaller than the other terms, which means that the saturation effect introduces but a small correction to the approximate simpler model. This correction can be, in principle, calculated by using regular perturbation theory, but respective biophysical insight from such exercise would be limited. We note, however, that such correction would have a significant effect for significantly higher concentrations of free ADF/Cofilin than those used in our experiments.

Quantitative estimates in **Mogilner and Edelstein-Keshet (2002)** show that in the case when profilin concentration is higher than both G-actin and ADF/Cofilin concentrations (in our case, there are 18 $\mu M$ of profilin, 6 $\mu M$ of G-actin, and less than 1 $\mu M$ of ADF/Cofilin), a major part of G-actin is associated with ATP, while ADF/Cofilin has a very low affinity to ATP-G-actin. More specifically, at these concentrations, there is a sub-$\mu M$ concentration of ADP-G-actin, only a few per cent of which is associated with ADF/Cofilin (most of the rest is associated with profilin), and so the order of magnitude of ADF/Cofilin concentration bound to G-actin is $\approx 10$ nM, which is less than 10% of the total ADF/Cofilin concentration. Thus, binding of ADF/Cofilin to G-actin can be neglected.

## Simulation

Parameters are summarized in **Appendix 1—Table 1** To solve **Equations (10), (11)** numerically, we used a splitting scheme: At each timestep $t_n$, we first solved the equations

$$\partial_t C_B = r_B A C_F - r_U C_B,$$

$$\partial_t C_F = D \Delta C_F - r_B A C_F + r_U C_B,$$

on the interval $[t_n, t_n + \Delta t]$ using a Finite Element Method (FEM) as implemented by the parabolic solver of Matlab's PDE toolbox. Since large density gradients can be expected only near the network, we used a triangular FEM-mesh that is much finer on and near the network than far away from it. Overall, the number of mesh triangles was between 2000 and 5000 for each simulation. The mesh itself was time-independent, avoiding having to re-mesh at each time step. Next, we employed a forward-Euler finite-difference scheme for the transport term $\partial_t C_B + V \partial_y C_B$, with shifting the calculated values of $C_B$ and subsequent interpolation onto the elements of the mesh.

## Stochastic fragmentation model: details and simulation

In our discrete network model, we describe the network as a collection of nodes and edges in 2D. Each edge represents an array of actin filaments, each note represents cross-linking or branching points. Our discrete network model follows ideas presented in **Carlsson (2007)**; **Michalski and Carlsson (2010)**, **Michalski and Carlsson (2011)**, however we allow our breakage rate to depend on local actin density and present a new analytical approximation (see Section Stochastic fragmentation model: details and simulation). At its fully connected

state each inner node is connected to four edges. We represent the whole network as a graph, that is for each node, we track the nodes to which this given node is connected to. During each time step, the discrete network model is updated in four steps:

1. *Remove individual nodes.* Given an actin density $A$ for a given node (step four below), we determine a breakage rate per node and time $P = \frac{p}{A^\alpha}$, where parameter $p$ is a constant. The node breakage follows a Poisson process with rate $P$, and we determine the probability of breakage at each time step as $1 - e^{-P\Delta t}$.

2. *Remove edges and network pieces.* There are two ways edges can be removed: Individually - this happens if both nodes an edge is connected to are removed. On the other hand, a larger network piece could become disconnected as a consequence of step 1. We considered network segments to be disconnected if they have no connection to the leading edge (i.e. there is no path of edges connecting the given piece to the leading edge) and assume disconnected network pieces diffuse away quickly.

3. *Grow network.* This step simply adds rows of nodes and edges at the leading edge proportional to the network growth speed $V$.

4. *Calculate local actin densities.* In our model, the local actin density depends on the number of edges present, not the number of nodes, that is if a node is removed within an otherwise fully connected patch, the actin density would not be affected. To calculate the local actin density at a node, we count the number of missing edges within a square patch around the node with a of $r_{\mathrm{nod}} = 2$ (in units of the edge length), that is we are considering the 24 nodes or 40 edges around the given node. Finally, the determined fraction of the unbroken edges was multiplied by the model parameter $A_0$, the initial actin density.

## Implementation and parameters

We performed numerical tests and found that as long as the node number along the leading edge, $K$, is larger than $\approx 20$, the equilibrium length is barely affected by the choice of $K$ (as noted also in **Carlsson (2007)**. We therefore decided to use $K = 30$, 60 and 180 for $15\mu m$, $30\mu m$ and $90\mu m$ networks, respectively. We used the time step of $\Delta t = 1$ min. For **Figure 4D** we varied the initial actin density $A_0$ and the exponent $\alpha$. For illustration purposes, we also changed the parameter $p$ and used $p = 0.25$, 2.5, 25 and 250 for $\alpha = 0.5$, 1, 1.5 and 2, respectively (otherwise the obtained network lengths would differ by orders of magnitudes, making the visualization less clear). For the simulation in main **Figure 4B** we used width = 90 μm, $\alpha = 2$, $A_0 = 100$ μM, $p = 194$ μM$^2$/min, $V = 0.8$ μm/min, for the comparison shown for the network in **Figure 4E** we used width = 15 μm, $\alpha = 2$, $A_0 = 400$ μM, $p = 250$ μM$^2$/min, $V = 1.5$ μm/min.

We represented the network as an undirected graph using Matlab routines, allowing to quickly determine connected components, which can be a time-consuming step. Edges that are connected to only one node can be represented as a loop, that is we formally connect both edge ends to the same node. Finally, if a node has two or three edges that are connected to only this node, this can be accounted for by assigning a weight of two or three to that edge. This is necessary to keep edges unique in the graph-based description. For example, a weight of two means that this edge counts twice when determining actin densities.

## Derivation of analytical model of network fragmentation

In this section, we describe an analytical approximation of the network length and actin density along the network of the discrete network model of Sec. Stochastic fragmentation model: details and simulation. In the discrete network model, we describe the whole network as a collection of nodes and edges, representing branches and connecting actin filaments, respectively. At its fully connected state each (inner) node is connected to four edges. Each nodes is being broken with a probability $P$ per node and time, that will depend on local properties of the network. An edge (i.e. actin filament segment) is removed only if either both of the nodes it is connected to are broken, or, if it is removed as part of a larger patch that is being disconnected.

## Main result

The analytical results can be summarized as follows: If the breakage rate per node is $P = c\frac{C_B^\beta}{A^\alpha}$, where $C_B$ is the (constant) concentration of bound ADF/Cofilin in the network, $A(y)$ is the actin density, $V$ is the network growth speed and $c$ is a dimensional constant proportionality coefficient, then the equilibrium network length is given by:

$$L = \frac{V A_0^\alpha}{c\, C_B^\beta} \int_0^1 \frac{(1-r^2)^\alpha}{(1-r)\left(1 + \frac{r}{(1-r)^2}\right)}\, dr.$$

Note that this integral is finite for any choice of $\alpha$. If $A_0$ is the initial actin density, then the actin density along the network in equilibrium is given by

$$A(y) = A_0(1 - r(y)^2),$$

where $r(y)$, the fraction of broken nodes along the network, is the solution of the ordinary differential equation:

$$V r' = c\frac{C_B^\beta}{A_0^\alpha}\frac{1-r}{(1-r^2)^\alpha}\left(1 + \frac{r}{(1-r)^2}\right), \quad r(0) = 1.$$

## Derivation

Let $N_0$ be the initial number of the network nodes per area, $R$ - the number of the broken nodes per area, and $E$ - the number of the network edges per area. If initially each node was connected to four edges, then $E_0 = 2N_0$. Two factors contribute to the edge removal:

A The current connectedness of the network;

B The number of nodes that are being removed locally.

A: In the absence of removal of larger pieces of the network, the deletion of a node will only affect edges that are connected to this very node, and only if those edges are unconnected at the other ends. The expected number of such edges for an unbroken node is:

$$\text{no. of edges per node} \times \text{prob. that the node at the other end is broken} = 4\frac{R}{N_0}.$$

We model continuous densities $R$ and $E$ using the following equations:

$$\dot{R} = P(N_0 - R)\left(1 + \frac{R/N_0}{(1 - R/N_0)^2}\right), \tag{12}$$

$$\dot{E} = -4\frac{R}{N_0}\dot{R}.$$

The second equation is simply stating that the rate of edge removal is equal to the rate of the node removal times the expected number of edges connected to the node being removed. In the first equation, expression $P(N_0 - R)$ accounts for the node breakage with rate $P$. Factor $\left(1 + \frac{R/N_0}{(1 - R/N_0)^2}\right)$ in this equation accounts for the factor B: if the network connectedness is low, then per each removed node, more nodes could be removed. This factor is equal to one for very low density of broken nodes, and has to be an increasing function of the variable $R/N_0$. Rather than using theoretical arguments to try to find this function, we simply used a few tens of simulations of the discrete stochastic model to estimate numerically the average number of the nodes removed for each randomly removed node at any given density of the broken nodes. The function $\left(1 + \frac{R/N_0}{(1 - R/N_0)^2}\right)$ approximated the numerical data well for $0 < R/N_0 < 0.7$. For larger $R/N_0$ the network is already largely falling

apart. We found that using more complicated functions to approximate the behavior hardly affects the predictions of network length and density.

## Adding transport effects

We introduce the space and time dependent fraction of broken nodes $r = R/N_0$ and the rescaled actin density $a = A/A_0$. Note that values of $A$ and $E$ are connected by the relation $A = E/\sqrt{N_0}$, where $1/\sqrt{N_0}$ is the approximate edge length in 2D. Since both edges and nodes are being transported within the network at speed $V$, we can replace **Equation (12)** by the following PDE system for densities $a(y,t)$ and $r(y,t)$, where $y$ is the distance along the network:

$$\partial_t r + V\partial_y r = P(1-r)\left(1 + \frac{r}{(1-r)^2}\right) =: P\rho(r), \tag{13}$$

$$\partial_t a + V\partial_y a = -2r\rho(r)P,$$

with the boundary conditions $a(0,t) = 1$, $r(0,t) = 0$.

## Explicitly calculating the network equilibrium length

In equilibrium, system (**Equation (13)**) takes the form:

$$Vr' = P(1-r)\left(1 + \frac{r}{(1-r)^2}\right) =: P\rho(r), \tag{14}$$

$$Va' = -2r\rho(r)P. \tag{15}$$

Rewriting **Equation (15)** as $a' = -(r^2)'$ shows that:

$$a(y) = 1 - r(y)^2. \tag{16}$$

Using the separations of variables, we can rewrite **Equation (14)** and find the following equation for the equilibrium length $L$:

$$V\int_0^{r(y)} \frac{1}{P\rho(\tilde{r})}\,d\tilde{r} = yV\int_0^1 \frac{1}{P\rho(r)}\,dr = L. \tag{17}$$

The final result depends on the choice of how the breakage rate $P$ depends on the actin density $A$ and the average amount of bound ADF/Cofilin $C_B$. We assume $C_B$ to be constant and use the breakage rate in the form:

$$P = c\frac{C_B^\beta}{A^\alpha} = c\frac{C_B^\beta}{(A_0 a)^\alpha} = c\frac{C_B^\beta}{A_0^\alpha}\frac{1}{(1-r^2)^\alpha},$$

where $c$ is the proportionality constant. This implies that the equilibrium length is given by:

$$L = \frac{V A_0^\alpha}{c\, C_B^\beta}\int_0^1 \frac{(1-r^2)^\alpha}{\rho(r)}\,dr.$$

The integral can be evaluated exactly:

$$L = \frac{V}{c}\frac{1}{C_B^\beta}\begin{cases} \frac{A_0}{2} & \alpha=1 \\ A_0^2\,\frac{27-4\sqrt{3}\pi}{12} & \alpha=2 \end{cases}$$

This is the formula used to compare the simulated equilibrium length to the calculated one in the main **Figure 4D**. Note that, given $P$, there are no free parameters, that is the

lengths are determined exactly. As described in the main text, we used the measured equilibrium lengths, actin densities and amounts of bound ADF/Cofilin to determine exponents $\alpha$ and $\beta$, and found good agreement for $\alpha = 2$, $\beta = 1$. This lead to main **Equation (5)** and is one of the ingredients used to calculate the equilibrium length below in Sec. Equilibrium lengths of homogeneous and heterogeneous networks.

## Equilibrium lengths of homogeneous and heterogeneous networks

### Homogeneous networks

First, we estimate the average amount of bound ADF/Cofilin $C_B$ in a network of a given length $L$. We use the estimate for the density of ADF/Cofilin in the solute in the vicinity of the network, derived in the main text (**Equation (3)**),

$$C_F \approx \frac{C_0 D + WLC_B r_U}{D + AWLr_B}.$$

Substitution of this expression into the equation for bound ADF/Cofilin (**Equation (10)**) yields:

$$\partial_t C_B + V \partial_y C_B = \chi(L) r_B A C_0 - \chi(L) r_U C_B,$$

$$C_B(y = 0) = 0,$$

where we define the depletion factor $\chi(L)$ as:

$$\chi(L) = \frac{D}{D + r_B A_0 L W}.$$

For a fixed network length $L$, this equation can be solved for any $y \leq Vt$:

$$C_B(x, y, t) = \frac{r_B A C_0}{r_U} \left(1 - e^{-\frac{r_U \chi(L) y}{V}}\right).$$

In our case, $r_U \approx 0.3 / \min$, $V \approx 1.5$ μm/min, $L \approx 30$ μm, $D \approx 600$ μm²/min, $A \approx 100$ μM, $r_B \approx 0.5 /$ min/μM, and so $\chi \approx 10^{-2}$. Hence we approximate the amount of bound ADF/Cofilin by the limit $r_U \to 0$, yielding:

$$C_B(x, y, t) = \frac{r_B A C_0 \chi(L) y}{V}.$$

The average amount of bound ADF/Cofilin, calculated as $\frac{1}{L} \int_0^L C_B(x, y, t) \, \mathrm{d}y$, is therefore given in **Equation (6)**:

$$C_B = \frac{r_B A_0 C_0 L}{2V} \chi(L). \tag{18}$$

From the network fragmentation model (see Sec. Stochastic fragmentation model: details and simulation) and the data, we found that for a given amount of bound ADF/Cofilin $C_B$, the network length is given by:

$$L = \kappa_S V \frac{(r_B A_0)^2}{C_B}, \tag{19}$$

where $\kappa_S = 0.067$ min³ × μM is the proportionality constant found by fitting to the data. Solving **Equations (18)-(19)** for $L$ and $C_B$ gives the equilibrium length $L^*$:

$$L^* = \frac{V}{C_0 D}\left( V W \kappa_S (r_B A_0)^2 + \sqrt{\left[ V W \kappa_S (r_B A_0)^2 \right]^2 + 2D^2 \kappa_S r_B A_0 C_0} \right).$$

This is the formula used for all equilibrium length predictions for **Figure 2**. Note that since the expression under the square root is always positive, the model predicts that the networks always reach an equilibrium length. Before the equilibrium length is reached, the length is simply given by $L = V t$, which explains the plateaus in **Figure 2B** and **Figure 2D**: According to the model, the networks had not yet reached equilibrium length at $t = 20$ min and $t = 38$ min.

## Heterogeneous networks

For heterogeneous networks we need to determine both equilibrium lengths $L_1$, $L_2$ for two sub-networks, that is we need to formulate **Equations (18) and (19)** separately for each sub-network. We denote by $A_{0,1}$ and $A_{0,2}$ the initial actin densities at the leading edge and by $V_1$ and $V_2$ the sub-network growth speeds. Since the networks are competing for the same pool of diffusing ADF/Cofilin, we assume that there is a common depletion factor, which we denote by $\chi_h(L_1, L_2)$ and model by the expression:

$$\chi_h(L_1, L_2) = \frac{D}{D + r_B W (A_{0,1} L_1 + A_{0,2} L_2)},$$

which takes into account the different densities and network lengths. Now we replace **Equation (18)** by:

$$C_{B,1} = \frac{r_B A_{0,1} C_0 L_1}{2 V_1} \chi_h(L_1, L_2), \quad C_{B,2} = \frac{r_B A_{0,2} C_0 L_2}{2 V_2} \chi_h(L_1, L_2), \tag{20}$$

and **Equation (19)** by:

$$L_1 = \kappa_S V_1 \frac{(r_B A_{0,1})^2}{C_{B,1}}, \quad L_2 = \kappa_S V_2 \frac{(r_B A_{0,2})^2}{C_{B,2}}. \tag{21}$$

All that remains is to solve the four **Equations (20)-(21)** for $L_1$, $L_2$, $C_{B,1}$ and $C_{B,2}$. We find the relation

$$\frac{L_1}{V_1 \sqrt{A_{0,1}}} = \frac{L_2}{V_2 \sqrt{A_{0,2}}},$$

which helps to simplify the equations. The final result can be written as:

$$L_1^* = \frac{V_1}{C_0 D}\left( V_1 W \kappa_S (r_B \tilde{A}_{12})^2 + \sqrt{\left[ V_1 W \kappa_S (r_B \tilde{A}_{12})^2 \right]^2 + 2D^2 \kappa_S r_B A_{0,1} C_0} \right),$$

$$L_2^* = \frac{V_2}{C_0 D}\left( V_2 W \kappa_S (r_B \tilde{A}_{21})^2 + \sqrt{\left[ V_2 W \kappa_S (r_B \tilde{A}_{21})^2 \right]^2 + 2D^2 \kappa_S r_B A_{0,2} C_0} \right),$$

where we have defined the terms $\tilde{A}_{ij}$ as:

$$\tilde{A}_{ij}^2 = A_{0,i}^2 + \frac{V_j}{V_i} A_{0,j} \sqrt{A_{0,i} A_{0,j}}.$$

The expressions for the equilibrium lengths are very similar to the homogeneous network case - in fact if $V_1 = V_2$ and $A_{0,1} = A_{0,2}$, they simplify to the case of one single network with width $2W$.

# Modeling the shape of heterogeneous networks

## Modeling

In this section, we model the shape of a network consisting of two sub-networks having different actin densities and/or growth speeds. We start by assuming that their material properties are constant along the network. We denote by $V_1 > V_2$ the growth speeds of sub-networks 1 and 2, respectively. As the network assembles, two sub-networks are effectively 'glued' together. In our simple model, we assume the sub-networks to be elastic. We model each sub-network segment by two springs, placed at a distance $W$ (the network width) from each other (*Appendix 1—figure 2*). The springs at the interface $\Gamma$ between the sub-networks are connected and forced to have the same length (representing the 'glued' together condition). The differences in sub-network growth speeds lead to different resting lengths $l_1$ and $l_2$, proportional to the respective speeds. To account for elastic effects, we assume that the differences in density lead to different elastic moduli $E_1$ and $E_2$, and hence to different spring constants $k_1 \propto E_1/l_1$ and $k_2 \propto E_2/l_2$. We call the length of the outermost and innermost spring $L_1$ and $L_2$ respectively; the length of the two springs at the interface therefore has to be $(L_1 + L_2)/2$.

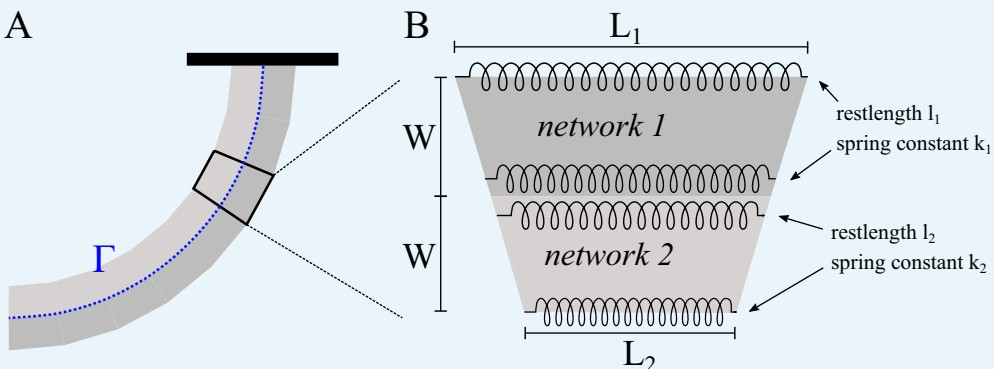

**Appendix 1—figure 2.** Modeling of heterogeneous, elastic networks. (**A**) Schematic of the experimental set-up. Different shading is used for the two sub-networks. The curve $\Gamma$ represents their interface. (**B**) Spring-based model of one network segment.
DOI: https://doi.org/10.7554/eLife.42413.022

## Results

To obtain the equilibrium lengths of the sub-networks, we minimize the elastic energy of each segment, which is given by adding the potential energies for each of the four springs:

$$E_{\text{pot}} = \frac{k_1}{2}\left[(L_1 - l_1)^2 + \left(\frac{L_1 + L_2}{2} - l_1\right)^2\right] + \frac{k_2}{2}\left[(L_2 - l_2)^2 + \left(\frac{L_1 + L_2}{2} - l_2\right)^2\right].$$

Minimization with respect to $L_1$ and $L_2$ gives:

$$L_1 = l_1 + \frac{k_2(k_1 - k_2)}{\delta}(l_1 - l_2), \quad L_2 = l_2 + \frac{k_1(k_1 - k_2)}{\delta}(l_1 - l_2),$$

where $\delta = k_1^2 + 6k_1 k_2 + k_2^2$. Using the basic proportionality theorem of elementary geometry of triangles, we find that the radius of curvature $R$ of the interface curve $\Gamma$ has to be:

$$R = W\frac{L_1 + L_2}{L_1 - L_2} = W\left(\frac{l_1 + l_2}{l_1 - l_2} + \frac{(k_1 - k_2)(k_1 l_1 - k_2 l_2)}{4k_1 k_2(l_1 - l_2)}\right).$$

To simplify interpretation, we denote by $\sigma = E_2/E_1$ the ratio of the elastic moduli and we use expressions: $l_1 = V_1\Delta t$, $l_2 = V_2\Delta t$, where $\Delta t$ is the time step. This gives:

$$R = W\left(\frac{V_1 + V_2}{V_1 - V_2} + (\sigma - 1)\frac{V_1\sigma - V_2}{4\sigma(V_1 - V_2)}\right).$$

The speed of the interface is given by:

$$V_h = \frac{L_1 + L_2}{2\Delta t} = \frac{V_1 + V_2}{2} - \frac{(V_1 - V_2)(V_1^2\sigma^2 - V_2^2)}{2(V_1^2\sigma^2 + 6V_1V_2\sigma + V_2^2)}.$$

These are the formulas used to calculate the curves shown in **Figure 6B,C** in the main text. Note that:

$$\infty \overset{\sigma\to 0}{\longleftarrow} R \overset{\sigma\to\infty}{\longrightarrow} \infty$$

$$V_1 \overset{\sigma\to 0}{\longleftarrow} v_h \overset{\sigma\to\infty}{\longrightarrow} V_2.$$

In other words, if one of the sub-networks is much stiffer that the other one, the heterogeneous network will become straight and grow with the speed of the stiffer sub-network.

## The density-elasticity scaling

In the formula for the curvature radius (**Equation (7)**) derived above, one needs to know the ratio of network elasticities $\sigma = E_2/E_1$. In the literature, scaling laws between elasticity $E$ and density $A$ of the form $E \propto A^\tau$ have been proposed with exponents varying from $\tau = 0.5$ (**Bieling et al., 2016**) to $\tau = 1.4$ (**Hinner et al., 1998**) to $\tau = 2.2$ (**Gardel et al., 2004b**) to $\tau = 2.5$ (**Gardel et al., 2004a**). We measured the actin network densities for the heterogeneous networks without ADF/Cofilin in **Figure 6D** and found an average density ratio of $0.78 \pm 0.04$ (mean±std) shown in **Appendix 1—figure 3A-B** and a curvature radius of $R = 75.3 \pm 17.9$ (mean± **Figure 6F**. Using these numbers in **Equation (7)** we can express the network speed ratio $V_2/V_1$ as a function of the exponent $\tau$, the resulting curve is shown in **Appendix 1—figure 3C**. We found that the speed ratio depends only mildly on $\tau$ and is roughly $V_2/V_1 = 2/3$. While we cannot measure the sub-network speed directly, we expect the ratio to be similar to the ratio of speeds of individually growing networks of corresponding patterns. Indeed, using the measurements from **Figure 1C** for low and high density patterns we find a ratio of $\approx 0.56$, which is in the correct range. In all simulations shown in **Figure 6** we used $\tau = 2.5$.

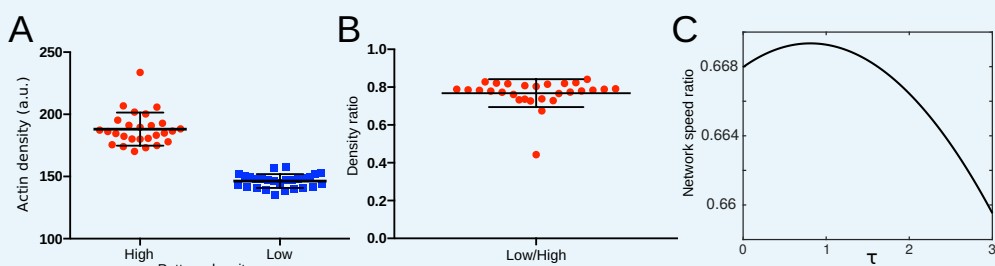

**Appendix 1—figure 3.** Density-elasticity scaling. (**A**) Variation of the actin density in heterogeneous patterns in **Figure 6D**, control. (**B**) Actin density ratio in heterogeneous patterns generated by the low and high density patterns in **Figure 6D**, control. (**C**) Showing the calculated ratio of sub-network speeds $V_2/V_1$ as function of the scaling exponent $\tau$. The relation is obtained from **Equation (7)** using an actin density ratio of 0.78 as measured in B as well as a curvature radius of 75 μm as measured in **Figure 6F**.
DOI: https://doi.org/10.7554/eLife.42413.023

## Sub-network shapes

Since our discussion so far concerned local properties of the sub-networks, we can account for changes in density along the combined network simply by making the elasticity ratio $\sigma$, and hence the curvature $\kappa$, a function of arc length along the network $s$. Then, curve $\Gamma(s)$ at the interface of the two sub-networks can be parametrized as $s \mapsto \Gamma(s) = (\Gamma_1(s), \Gamma_2(s))$ (*Appendix 1—figure 2*), where:

$$\Gamma_1(s) = \int_0^y \sin\left(\int_0^z \kappa(w)\,dw\right)dz, \quad \Gamma_2(s) = \int_0^y \cos\left(\int_0^z \kappa(w)\,dw\right)dz.$$

In case of constant material properties along each sub-network (as shown in *Figure 6A* in the main text), these expressions simplify to:

$$\Gamma_1(s) = \frac{1 - \cos(\kappa s)}{\kappa}, \quad \Gamma_2(s) = \frac{\sin(\kappa s)}{\kappa}.$$

If actin densities vary along the sub-network, we can use the analytical approximation of the discrete model for network fragmentation described in Sec. Stochastic fragmentation model: details and simulation. In particular, we have to solve *Equation (14)* to obtain the density of broken nodes, after which we can use *Equation (16)* to obtain a formula for the actin density along the sub-network. To simplify the results, we use the following approximation of *Equation (14)*:

$$Vr' = c\frac{C_B}{A_0^2}\frac{1}{(1-r)^2}. \tag{22}$$

Using the notation introduced above in Sec. Equilibrium lengths of homogeneous and heterogeneous networks and the calculated equilibrium lengths $L_1^*$ and $L_2^*$, we obtain:

$$r_1(s) = 1 - (1 - s/L_1^*)^{1/3}, \quad A_1(s) = A_{0,1}(1 - r_1^2(s)), \quad s < L_1^*$$

$$r_2(s) = 1 - (1 - s/L_2^*)^{1/3}, \quad A_2(s) = A_{0,2}(1 - r_2^2(s)), \quad s < L_2^*.$$

Since $L_1^* > L_2^*$, we can approximate the elasticity ratio of the two sub-networks as:

$$\sigma(s) = \begin{cases} \left(\frac{A_2(s)}{A_1(s)}\right)^{2.5} & y \in [0, L2^*], \\ \infty & \text{elsewise.} \end{cases} \tag{23}$$

These are the formulas used to calculate the network shapes shown in *Figure 6E* in the main text. As parameters, we used the density measurements from the ADF/Cofilin free case shown in *Appendix 1—figure 3A* for $A_{0,1}$ and $A_{0,2}$ and the speeds $V_1 = 2$ μm/min and $V_2 = 1.33$ μm/min. Note that in the figure, networks have been rotated to match the experimental set-up.

