## [Decision Letter]

Thank you for submitting your article "Reconstitution of the equilibrium state of dynamic actin networks" for consideration by *eLife*. Your article has been reviewed by three peer reviewers, including Pekka Lappalainen as the Reviewing Editor and Reviewer #1, and the evaluation has been overseen by Anna Akhmanova as the Senior Editor. The following individual involved in review of your submission has also agreed to reveal their identity: R Dyche Mullins (Reviewer #3).

The reviewers have discussed the reviews with one another and the Reviewing Editor has drafted this decision to help you prepare a revised submission.

Summary:

The effects of various actin-binding proteins on the dynamics of individual actin filaments have been extensively studied, but much less in known about how they regulate physiological networks of actin filaments. Earlier study by Boujemaa-Paterski et al., 2017, revealed that both width and density of the Arp2/3-nucleated networks affect their growth rate and steering in vitro. Here, Manhart et al., applied similar in vitro reconstitution and mathematical modeling approaches to study the effects of ADF/cofilin on the network dynamics. Most importantly, they reveal that: i). ADF/cofilin disintegrates the actin network into micron size fragments, ii). Local depletion of ADF/cofilin by binding to actin filaments results in wider networks growing longer, and iii). ADF/cofilin can control steering of heterogenous branched actin networks.

Although this study is an "extension" of a previous work, all three reviewers found it novel and interesting, and addressing issues that are often brought up but never directly examined, especially not in controlled conditions like here. However, the reviewers identified several important points that need to be addressed in the revised version of the manuscript. These are significant issues, but many of them are conceptual and could be handled without new experiments.

Essential revisions:

1) It seems that, as filaments become decorated by cofilin (eventually reaching saturation, or at least a steady-state level of decoration) the amount of F-actin available to drive further depletion decreases. This seems essential but is never taken into account, nor discussed. In Equations 1 and 2, the binding of cofilin stays proportional to A, which is the total (including already decorated) actin filament density. It is not clear that certain conclusions would hold without this extreme simplification. For instance, beyond a certain network length, the filaments would be saturated and no longer contribute to cofilin depletion. This issue should be thoroughly addressed.

2) The in vitro experiments were performed by using very thin experimental chambers to make the networks flat, similar to the lamellipodial actin filament networks. The 'lengths' of the reconstituted (lamellipodial) actin networks e.g. in Figure 2C are 50 – 90 um, whereas the 'length' of a lamellipodium in most migrating cells, apart from fish keratinocytes, is approximately 1-4 um. Moreover, the sizes of fragments disintegrating from the network (e.g. Figure 4) are often larger than the 'length' of a typical lamellipodium of a migrating cell. Thus, the authors should better discuss whether local depletion of ADF/cofilin by binding to actin filament networks can indeed have major effects on the disassembly/geometry of similar (lamellipodial) actin filament networks also in cells. Moreover, they should discuss whether similar stochastic fragmentation of actin filament networks into small pieces can also occur in cellular actin filament structures (which are much smaller in size, and where ADF/cofilin cooperates with many 'co-factors' such as Aip1 and Srv2/CAP in actin filament disassembly).

3) In their experiments, the authors used yeast cofilin and rabbit muscle actin. We know that this will be a lot less efficient at decorating and severing filaments than if they had used mammalian cofilin. This may explain why such long networks were observed. Moreover, the authors may be in a regime where the filaments are mostly far from cofilin saturation, which would partially validate the crude approximation they make regarding the depletion of cofilin (neglecting the decrease in the number of available sites for cofilin binding on actin filaments; see point 1 above). Therefore, this aspect should be discussed.

4) There are some elements of the mathematics that don't quite make sense. For example, Equation 7 describes the network curvature, using a parameter, "kappa", which is usually defined as the inverse of the radius of curvature. The units of kappa should, therefore, be 1/µm. According to Equation 7, however, the units for kappa are µm. This issue also appears in Figure 6B, where the y-axis is labeled "Curvature Radius" but the units of the dashed line are clearly 1/µm.

More generally, the mathematical modeling assumes that stiffness varies as A^2.5 (subsection “ADF/Cofilin regulates steering of heterogeneous networks”, second paragraph), a result that is derived from the study of random, entangled or crosslinked actin networks. Others have reported a very different concentration dependence for branched actin networks assembled under the sort of boundary conditions used here, namely A^0.5. This much weaker dependence on concentration is consistent with the assumptions made in the previous paper (Boujemaa-Paterski, 2017), and mentioned in the aforementioned paragraph. The authors have the opportunity to use their curvature measurements to independently judge the concentration dependence of the stiffness of branched actin networks.

Minor points follow.

*Reviewer #1:*

1) The authors state in the 'Discussion' that by using their formulas they can estimate the lamellipodial length in motile cells of intermediate size. Given that their equations do not include other actin filament disassembly factors (e.g. Aip1, coronin, Srv2/cofilin, twinfilin), and other F-actin binding proteins that affect filament disassembly (tropomyosins, filament cross-linking proteins), this does not seem feasible. Therefore, the authors should either omit this part of the discussion or alternatively provide better explanation for how these additional factors were taken into account in these estimations.

2) Many figures are quite complex and thus somewhat difficult to follow. For example, Figure 3 would benefit if the authors would try to simplify it (e.g. by moving any 'non-essential' panels to supplementary information, and by providing more information about each panel in the legend). Moreover, especially the chapters 'Rate of ADF/cofilin binding decreases with time' and 'ADF/cofilin is locally depleted by binding to the growing actin network' would benefit from extensive rewriting to make the conclusions better accessible to a 'non-specialist' reader.

3) ADF/cofilin binds both F-actin and G-actin. Was this taken into account in simulations performed in this study?

4) Definition of the error bar is missing from the legend to Figure 4A.

*Reviewer #2:*

1) In the simulations (Figure 3D) the amount of free ADF/cofilin in the network decreases only. Is that because the model assumes that the number of binding sites on actin remains constant (see essential point 1)?

2) Subsection “ADF/Cofilin is locally depleted by binding to the growing actin network”, third paragraph; Depletion is computed by considering the global balance over the whole network, in order to describe what happens near the leading edge. This cannot be right, since there are important local variations.

3) In Figure 3A, the initial slopes decrease, but the green curve also seems to plateau much lower than the blue curve. How can that be?

4) Since this work deals with the establishment of steady-state, this aspect should be documented a bit more. The authors mention the increase of the network length, which goes on regularly without ADF/cofilin but reaches and equilibrium length in the presence of ADF/cofilin. The authors should show measurements of L as a function of time.

5) The experimental observation regarding the actin profile, which is relatively constant and plunges only near the trailing edge is somewhat surprising. It can be reproduced by the model, which (as the authors clearly state) crudely simplifies the action of cofilin: network nodes are removed beyond a certain threshold of decoration. The authors discuss (…subsection “Relation to previous studies”, second paragraph) that this is consistent with previous reports on debranching at low cofilin concentrations, however it is well established in the literature that the maximum severing is reached for intermediate levels of cofilin decoration. Also, I believe previous reports on the actin-based propulsion of beads or droplets generally find a more progressive decrease in actin density. This should be discussed more.

6) Some statements in the text are exaggerated: e.g., "this provides a direct demonstration…" (a model matching experimental data does not provide a direct demonstration that the hypothesis is what is actually going on), or, "we made the novel observation that heterogeneous networks grow curved" (already reported in Boujemaa-Paterski, 2017).

*Reviewer #3:*

1) The title is misleading and should be changed. Firstly, "equilibrium" suggests that the authors are studying a thermodynamic equilibrium rather than a quasi-steady state network treadmilling that carries on for as long as the ATP in the system remains high. Secondly, the title suggests that the novelty of the paper lies in the "reconstitution" of steady-state actin network turnover. As the authors note, this has been reconstituted many times before. The novelty lies in the quantitative approach and some of the mechanistic details that emerge from it.

2) I cannot find any information on the nucleation promoting factor that is patterned on the glass surface and used to generate branched actin networks in this study. In the text and figure captions the molecule is simply described as "an NPF." I don't see it in the Materials and methods either. It might be buried somewhere in the manuscript, but it should be prominent in text and noted in all of the captions for figures containing experimental data. This is important for several reasons: (1) different NPFs have different cellular functions and stimulate different rates of nucleation and polymerization, and (2) it is unclear which domains are in the construct (full-length protein? PWCA? WCA?). In a previous paper (Bougemaa-Paterski, 2017) the authors described their immobilized NPF construct as a GST-pWCA. If this is the same construct, the dimeric nature of the GST would also be significant.

3) Figure 1 demonstrates that micro-patterns with different densities of (an unknown construct of an unknown) NPF promote different velocities of network growth. The result is striking and believable but should be described in more detail. For example, what are the relative densities of the micro-patterned NPFs corresponding to "low," "medium," and "high" densities?

4) On a related note, I disagree with the authors' explanation for the correlation between network density and velocity. They state that: "higher NPF density that causes greater actin density and a moderate depletion of monomeric actin in the vicinity of the growing barbed ends, also leads to an optimization of the micro-architecture of the network, which translates polymerization into the network elongation more effectively for denser networks". I see no evidence to support this claim in either this paper or their previous work (Boujemaa-Paterski, 2017). My (admittedly biased) interpretation is based on the fact that we recently demonstrated that NPF's have a potent polymerase activity that accelerates elongation of nearby filaments. This results in a growth velocity that is directly proportional to the NPF surface density.

---

## [Author Response]

Essential revisions:1) It seems that, as filaments become decorated by cofilin (eventually reaching saturation, or at least a steady-state level of decoration) the amount of F-actin available to drive further depletion decreases. This seems essential but is never taken into account, nor discussed. In Equations 1 and 2, the binding of cofilin stays proportional to A, which is the total (including already decorated) actin filament density. It is not clear that certain conclusions would hold without this extreme simplification. For instance, beyond a certain network length, the filaments would be saturated and no longer contribute to cofilin depletion. This issue should be thoroughly addressed.

We appreciate the insightful comment. We have addressed the issue in the new subsection called “Actin saturation” in the Appendix. Here we formulate a more detailed model in which we distinguish between decorated and free F-actin, where only the latter is available for the binding of ADF/Cofilin. This can be seen as a correction of the model in the main text where the total Factin is available for binding. By examining the relative magnitudes of the different terms, we found that this correction is in fact an order of magnitude smaller than the other terms at the concentrations of ADF/Cofilin used in our experiments. Thus, for our data, saturation of the actin filaments with cofilin plays a negligible role justifying the use the simplified model presented in the main text.

2) The in vitro experiments were performed by using very thin experimental chambers to make the networks flat, similar to the lamellipodial actin filament networks. The 'lengths' of the reconstituted (lamellipodial) actin networks e.g. in Figure 2C are 50 – 90 um, whereas the 'length' of a lamellipodium in most migrating cells, apart from fish keratinocytes, is approximately 1-4 um. Moreover, the sizes of fragments disintegrating from the network (e.g. Figure 4) are often larger than the 'length' of a typical lamellipodium of a migrating cell. Thus, the authors should better discuss whether local depletion of ADF/cofilin by binding to actin filament networks can indeed have major effects on the disassembly/geometry of similar (lamellipodial) actin filament networks also in cells. Moreover, they should discuss whether similar stochastic fragmentation of actin filament networks into small pieces can also occur in cellular actin filament structures (which are much smaller in size, and where ADF/cofilin cooperates with many 'co-factors' such as Aip1 and Srv2/CAP in actin filament disassembly).

Let us first note that, depending on the conditions, we do see very short – on the order of 10 micron long – networks, which are comparable to the length of the keratocyte lamellipodia. Second, we have estimated the effect of local depletion on actin tails and lamellipodia-like structures in the Discussion, and there is a clear prediction of the depletion effect. Perhaps these estimates were not well organized; we now organize them better in the section ‘Novelty of our findings and relevance to in vivo networks’. Even for very short lamellipodia, the depletion effect is there.

Stochastic fragmentation: Directly, the ‘macroscopic’ fragmentation was only observed in in vitro reconstitution experiments (Reymann et al., 2011). It would be indeed hard to imagine breakage of microns-size fragments from 1-4 um-long lamellipodia. However, it is not out of the question that such fragmentation could take place at the trailing edge of longer, ~ 10 μm long, lamellipodia. The problem is that it is hard to imagine how such process could be imaged in vivo. This is the problem for the future. Note though that some of the images in the literature – consider, for example, Figure 1 of (Cameron et al., 1999) – clearly show patchy actin comet tails, which could be interpreted as the disassembly through fragmentation. Another point is that ‘microscopic’ fragmentation through breaking actin filaments into small pieces was inferred in a few studies. How disassembly co-factors change the fragmentation scenario is a great question for the future. One possibility is that these co-factors make the disassembly smoother: it was demonstrated that concerted action of Cofilin, Aip1 and Coronin first breaks filaments into small fragments, and then disassembles the fragments into monomers at such speed that effectively the disassembly is continuous (Johnston et al., 2015).

We revised relevant parts of the Discussion accordingly.

3) In their experiments, the authors used yeast cofilin and rabbit muscle actin. We know that this will be a lot less efficient at decorating and severing filaments than if they had used mammalian cofilin. This may explain why such long networks were observed. Moreover, the authors may be in a regime where the filaments are mostly far from cofilin saturation, which would partially validate the crude approximation they make regarding the depletion of cofilin (neglecting the decrease in the number of available sites for cofilin binding on actin filaments; see point 1 above). Therefore, this aspect should be discussed.

We thank the reviewers for pointing out this possibility. We previously determined the Kd of yeast cofilin for muscle actin to be below 300 nM (Suarez et al., 2011) where the Kd of human cofilin is between 3 and 4 µM (published recently in Chin, Jansen and Goode, 2016). For the debranching activity, probably the most relevant activity for this study, yeast cofilin was at least as efficient as human cofilin, so, while this is an interesting possibility, we do not think that the long networks observed in some of the condition are due to the use of yeast cofilin. In any case, we revised parts of the Discussion to clarify these points.

4) There are some elements of the mathematics that don't quite make sense. For example, Equation 7 describes the network curvature, using a parameter, "kappa", which is usually defined as the inverse of the radius of curvature. The units of kappa should, therefore, be 1/µm. According to Equation 7, however, the units for kappa are µm. This issue also appears in Figure 6B, where the y-axis is labeled "Curvature Radius" but the units of the dashed line are clearly 1/µm.

We thank the reviewer for pointing out the typos. We corrected them both in the text and in the figures. Now the dimension of the radius of curvature is µm, as it should be.

More generally, the mathematical modeling assumes that stiffness varies as A^2.5 (subsection “ADF/Cofilin regulates steering of heterogeneous networks”, second paragraph), a result that is derived from the study of random, entangled or crosslinked actin networks. Others have reported a very different concentration dependence for branched actin networks assembled under the sort of boundary conditions used here, namely A^0.5. This much weaker dependence on concentration is consistent with the assumptions made in the previous paper (Boujemaa-Paterski, 2017), and mentioned in the aforementioned paragraph. The authors have the opportunity to use their curvature measurements to independently judge the concentration dependence of the stiffness of branched actin networks.

This is a very good point, and we discuss it in detail in the new subsection “The density-elasticity scaling” in the Appendix. We measured the densities of the sub-networks for heterogeneous networks in the control case (Appendix 1—figure 3) and used the measurements to calculate the dependence of the ratio of the growth speeds of the sub-networks on the scaling exponent τ in the relation ∝𝐴. We made two observations: 1) the speed ratio depends only very mildly on the value of τ, and 2) the predicted speed ratio is in the correct range. Observation 1 shows that one would have to measure the speed ratio with very high accuracy to calculate τ from the data, which is not possible with the data at hand.

When reviewing the literature, we could not find report(s) of τ = 0.5, but we found reported values ranging from 1.4 to 2.5 (we added references to respective studies). As we said (and added the data in the Appendix), the results depend very little on the value of τ, and even for the value of 0.5. The model agrees well with the data.

Minor points follow.Reviewer #1:1) The authors state in the 'Discussion' that by using their formulas they can estimate the lamellipodial length in motile cells of intermediate size. Given that their equations do not include other actin filament disassembly factors (e.g. Aip1, coronin, Srv2/cofilin, twinfilin), and other F-actin binding proteins that affect filament disassembly (tropomyosins, filament cross-linking proteins), this does not seem feasible. Therefore, the authors should either omit this part of the discussion or alternatively provide better explanation for how these additional factors were taken into account in these estimations.

We realized, thanks to the reviewer, that parts of the Discussion were not clear on this point. We are not, of course, proposing that we can calculate the length of the actin networks in cells exactly. What we do is extrapolating our findings on in vivo situation and looking at respective predictions. The results (limited even for just ADF/Cofilin, without other factors, because the action of ADF/Cofilin does not scale linearly with its concentration) give reasonable estimates of the networks’ lengths, which are still longer than usually observed networks in vivo. This estimate points exactly to the importance of the additional disassembly co-factors, as the reviewer points out. For example, we predict the network length of tens of microns, which agrees with in vitro reconstitutions (Loisel, 2000; Reymann, 2011), but is longer than a few micronslong lamellipodia in vivo and micron-sized yeast endocytic networks. There are some indications from studies of Goode’s lab that effects of the disassembly co-factors accelerate the disassembly by a few-fold to an order of magnitude, which would bring our estimates to the observed range. We revised the Discussion accordingly.

2) Many figures are quite complex and thus somewhat difficult to follow. For example, Figure 3 would benefit if the authors would try to simplify it (e.g. by moving any 'non-essential' panels to supplementary information, and by providing more information about each panel in the legend). Moreover, especially the chapters 'Rate of ADF/cofilin binding decreases with time' and 'ADF/cofilin is locally depleted by binding to the growing actin network' would benefit from extensive rewriting to make the conclusions better accessible to a 'non-specialist' reader.

We moved parts of Figure 3 to a new supplementary figure and improved the description in the captions. We also split up the very busy Figure 5 into two separate Figures 5 and 7.

We also re-wrote the chapters 'Rate of ADF/cofilin binding decreases with time' and 'ADF/cofilin is locally depleted by binding to the growing actin network' to make them better accessible.

3) ADF/cofilin binds both F-actin and G-actin. Was this taken into account in simulations performed in this study?

This is an important question, which we neglected to discuss; we are grateful to the reviewer for drawing attention to it. Quantitative estimates in our study (Mogilner and Edelstein-Keshet, 2002) show that in the case when profilin concentration is higher than both G-actin and ADF/Cofilin concentrations (in our case, there are 18 μM of profilin, 6 μM of G-actin, and less than1 mM of ADF/Cofilin), a major part of G-actin is associated with ATP, while ADF/Cofilin has a very low affinity to ATP-G-actin. More specifically, at these concentrations, there is a sub-μM concentration of ADP-G-actin, only a few per cent of which is associated with ADF/Cofilin (most of the rest is associated with profilin), and so the order of magnitude of ADF/Cofilin concentration bound to G-actin is ~ 10 nM, which is less than 10% of the total ADF/Cofilin concentration. The bottom line is that binding of ADF/Cofilin to G-actin can be neglected. We added a few sentences to this effect in the Appendix.

4) Definition of the error bar is missing from the legend to Figure 4A.

We added the definition in the figure legend.

Reviewer #2:

1) In the simulations (Figure 3D) the amount of free ADF/cofilin in the network decreases only. Is that because the model assumes that the number of binding sites on actin remains constant (see essential point 1)?

As we explain above, decoration of some F-actin with ADF/Cofilin has only small, negligible, effect on the results. Rather, the decrease of the free ADF/cofilin in the immediate vicinity of the network with time is due to the local ADF/cofilin depletion. Qualitatively, the explanation is as follows: initially, when the network is small, ADF/cofilin starts binding to F-actin starting to decrease the local ADF/cofilin concentration. Diffusion brings more ADF/cofilin to the network, and the balance between ADF/cofilin binding to F-actin and diffusion influx define the local ADF/cofilin concentration. As the network keeps growing, the sink for the free ADF/cofilin increases, further decreasing the local free ADF/cofilin concentration. Eventually, this decrease of the local free ADF/cofilin concentration slows down because the older part of the actin network binds ADF/cofilin at lower rates.

2) Subsection “ADF/Cofilin is locally depleted by binding to the growing actin network”, third paragraph; Depletion is computed by considering the global balance over the whole network, in order to describe what happens near the leading edge. This cannot be right, since there are important local variations.

These calculations are not exact; they are by no means intended to replace the accurate numerical calculation of the cofilin densities, which we do numerically. These calculations are rough estimates of the average free cofilin concentration in the vicinity of the actin network. These estimates indeed ignore the local variations of the cofilin concentration (the latter are clearly seen in the accurate numerical results shown in the figures); however, because of diffusion, the local variations of the cofilin concentration is relatively limited (numerical calculation demonstrates that the s.t.d. of the free cofilin concentration is about 35% of the mean). The rough estimate is very useful because it allows obtaining analytical insight about the magnitude of the depletion effect. We added a few words to the main text to explain this issue.

3) In Figure 3A, the initial slopes decrease, but the green curve also seems to plateau much lower than the blue curve. How can that be?

Figure 3A shows the bound ADF/cofilin density at a given material point of the actin network, starting from the appearance of such point at the leading edge. The ‘blue’ point is the first to appear, when the actin network is very short. Therefore, the local depletion effect is very weak, and the rate of ADF/cofilin binding is high. With time, the network grows, the depletion effect strengthens, and the rate of binding decreases; eventually, the bound ADF/cofilin density on the ‘blue’ point plateaus. The ‘green’ point appears later, when the depletion effect is already pronounced. Thus, the initial binding rate is lower, and over time the accumulation of the ADF/cofilin is less, hence the lower plateau. Perhaps origin of the confusion is this: the plateau is not due to the kinematic balance between the binding and unbinding of ADF/cofilin (the unbinding is very slow). Rather, the plateau is the accumulated, integrated over time, bound ADF/cofilin. Also, the plateau, on average, is not truly flat, as it appears in the examples in Figure 3A. The more characteristic situation is shown in the old Figure 3F (new Figure 3—figure supplement 1B).

4) Since this work deals with the establishment of steady-state, this aspect should be documented a bit more. The authors mention the increase of the network length, which goes on regularly without ADF/cofilin but reaches and equilibrium length in the presence of ADF/cofilin. The authors should show measurements of L as a function of time.

We prepared a new figure (Figure 2—figure supplement 1) documenting the growth behavior of the networks of different widths, with and without cofilin by measuring the network length as a function of time. This shows the dynamic steady state behavior in the presence of cofilin.

5) The experimental observation regarding the actin profile, which is relatively constant and plunges only near the trailing edge is somewhat surprising. It can be reproduced by the model, which (as the authors clearly state) crudely simplifies the action of cofilin: network nodes are removed beyond a certain threshold of decoration. The authors discuss (subsection “Relation to previous studies”, second paragraph) that this is consistent with previous reports on debranching at low cofilin concentrations, however it is well established in the literature that the maximum severing is reached for intermediate levels of cofilin decoration. Also, I believe previous reports on the actin-based propulsion of beads or droplets generally find a more progressive decrease in actin density. This should be discussed more.

Let us first note that in the model the nodes are removed not beyond a threshold, but with a rate smoothly dependent on the ADF/Cofilin and actin densities, so the nature of the observed abrupt actin density plunge is not abrupt node removal. Rather, it is that the first stage of the disassembly – node removal – leads to a slow density decrease, which later accelerates by the positive feedback between the actin drop and node disappearance. We are also not arguing with maximum severing being reached at intermediate levels of cofilin decoration. It could well be that at higher cofilin densities, and especially with additional disassembly co-factors, the density decrease is more gradual. Indeed, most of the previous measurements showed more gradual decrease of the actin density in the actin tails. However, in vitro reconstitutions (Loisel, 2000; Reymann, 2011) show the abrupt density decrease. Also, abrupt actin density decrease at the rear of the lamellipodia in keratocyte cells and fragments (Barnhart, 2011; Ofer, 2011) was observed. We revised the Discussion accordingly.

6) Some statements in the text are exaggerated: e.g., "this provides a direct demonstration…" (a model matching experimental data does not provide a direct demonstration that the hypothesis is what is actually going on), or, "we made the novel observation that heterogeneous networks grow curved" (already reported in Boujemaa-Paterski, 2017).

We toned down all such statements.

Reviewer #3:

1) The title is misleading and should be changed. Firstly, "equilibrium" suggests that the authors are studying a thermodynamic equilibrium rather than a quasi-steady state network treadmilling that carries on for as long as the ATP in the system remains high. Secondly, the title suggests that the novelty of the paper lies in the "reconstitution" of steady-state actin network turnover. As the authors note, this has been reconstituted many times before. The novelty lies in the quantitative approach and some of the mechanistic details that emerge from it.

We changed the title to: ‘Quantitative regulation of the dynamic steady state of actin networks’.

2) I cannot find any information on the nucleation promoting factor that is patterned on the glass surface and used to generate branched actin networks in this study. In the text and figure captions the molecule is simply described as "an NPF." I don't see it in the Materials and methods either. It might be buried somewhere in the manuscript, but it should be prominent in text and noted in all of the captions for figures containing experimental data. This is important for several reasons: (1) different NPFs have different cellular functions and stimulate different rates of nucleation and polymerization, and (2) it is unclear which domains are in the construct (full-length protein? PWCA? WCA?). In a previous paper (Bougemaa-Paterski, 2017) the authors described their immobilized NPF construct as a GST-pWCA. If this is the same construct, the dimeric nature of the GST would also be significant.

The NPFs were described in the Materials and methods sections. It is indeed the same WASp, GST-pVCA than in Boujemaa-Paterski et al., 2017. We use GST- construct because in previous study (Cory et al., Molecular Cell, 2003) we reported that the GST- construct had a similar activity than the WASp full length. In any case, we have defined as suggested by the reviewer the NPF in the text as Human WASp, GST-pWCA.

3) Figure 1 demonstrates that micro-patterns with different densities of (an unknown construct of an unknown) NPF promote different velocities of network growth. The result is striking and believable but should be described in more detail. For example, what are the relative densities of the micro-patterned NPFs corresponding to "low," "medium," and "high" densities?

This is illustrated in Figure 1—figure supplement 1C, where we have plotted the fluorescence of the WASp-GST-pWCA versus the pattern density Low, Medium, High.

4) On a related note, I disagree with the authors' explanation for the correlation between network density and velocity. They state that: "higher NPF density that causes greater actin density and a moderate depletion of monomeric actin in the vicinity of the growing barbed ends, also leads to an optimization of the micro-architecture of the network, which translates polymerization into the network elongation more effectively for denser networks". I see no evidence to support this claim in either this paper or their previous work (Boujemaa-Paterski, 2017). My (admittedly biased) interpretation is based on the fact that we recently demonstrated that NPF's have a potent polymerase activity that accelerates elongation of nearby filaments. This results in a growth velocity that is directly proportional to the NPF surface density.

To address this comment, we measured experimentally the rate of elongation of single actin filaments on the NPF patterns of different densities. We did not find any statistically significant difference in these rates for different NPF densities. For individual filaments, the rate of elongation is V = k_on*C*δ, where C is the local monomer concentration, and δ is the half-size of the monomer. Our previous estimates in (Boujemaa-Paterski, 2017) showed that for a small number of growing filaments, the local monomer concentration is not depleted, and so parameter C is independent of the NPF density. We conclude therefore that the rate of assembly for a single filament, k_on, is also independent of the NPF density. For the actin network, the rate of elongation is V = k_on*C*δ*Fi, where factor Fi combines potential geometric (angular order and entanglements of the leading network filaments) and mechanical (dependence of the force balance at the network edge and/or balance of actin growth and elastic recoil, both of which depend on density) effects. We observe that V goes up with density, while C was argued to be decreasing with density (in (Boujemaa-Paterski, 2017)), so the most logical conclusion would be that factor Fi increases with the NPF density. We added the data and this argument to the Appendix; in the main text we now write:

“In the appendix, we report data suggesting that higher NPF density that causes greater actin density could lead to an optimization of either micro-architecture or mechanics of the network, which translates polymerization into the network elongation more effectively for denser networks.”